# Decay channels for double extremal black holes in four dimensions

**Johannes Aspman**[a]**, Jan Manschot**[b,c,d]

[a]*Department of Computer Science, Czech Technical University in Prague, Czech Republic*
[b]*School of Mathematics, Trinity College, Dublin 2, Ireland*
[c]*Hamilton Mathematical Institute, Trinity College, Dublin 2, Ireland*
[d]*School of Natural Sciences, Institute for Advanced Study, 1 Einstein Drive, Princeton, NJ 08540 USA*

ABSTRACT: We explore decay channels for charged black holes with vanishing temperature in $\mathcal{N} = 2$ supersymmetric compactifications of string theory. If not protected by supersymmetry, such extremal black holes are expected to decay as a consequence of the weak gravity conjecture. We concentrate on double extremal, non-supersymmetric black holes for which the values of the scalar fields are constant throughout space-time, and explore decay channels for which decay into BPS and anti-BPS constituents is energetically favorable. We demonstrate the existence of decay channels at tree level for large families of double extremal black holes. For specific charges, we also find stable non-supersymmetric black holes, suggesting re-combination of (anti)-supersymmetric constituents to a non-supersymmetric object.

# 1   Introduction

Black holes with vanishing temperature do not emit Hawking radiation. Still such extremal black holes are expected to decay, if not protected by supersymmetry, as a consequence of the Weak Gravity Conjecture (WGC) [1]. This conjecture states that gravity is the weakest force in any consistent theory of quantum gravity. More precisely, the conjecture states that in any consistent theory of quantum gravity there must exist at least one object on which the gravitational force is smaller than that

due to a gauge charge [1–3]. Consequently, there must be a bound on the mass of this object in terms of the other charges of the theory. For example, in four-dimensional gravity with a $U(1)$ gauge field this bound reads [3]

$$M \leq M_p Q, \tag{1.1}$$

where $Q$ is the charge and $M_p$ the four-dimensional Planck mass. The stronger version of the conjecture, put forward in [4], states that only BPS black holes are allowed to saturate the weak gravity bound. It is important to note that the WGC is a statement about low-energy effective theories. As such, it does not make qualitative predictions on the exact UV completion of the theory. It is possible that non-supersymmetric states of the UV completion are stable against decay [29].

Decay channels of black holes have been studied in many cases, see for example [5–13]. A common approach is to consider higher derivative corrections, and to demonstrate that their coefficients imply that the ratio $M/Q$ decreases with increasing $Q$. In theories with gauge group $U(1)^b$, $b > 1$, decay of extremal black holes leads to further conditions, in particular conditions on the convex hull of charge-to-mass ratios [6].

In the present paper, we will consider decay channels for four-dimensional black holes in $\mathcal{N} = 2$ supergravity. These theories have generically multiple $U(1)$ gauge fields, as well as families of solutions of extremal black holes, both supersymmetric [14, 15][1] as well as non-supersymmetric [17–19]. Both families involve non-trivial dynamics of the scalar fields, known as the attractor mechanism, which describes the evolution of the scalar fields from asymptotic infinity to the horizon. As described above, the weak gravity conjecture suggests that it is energetically favorable for the extremal black holes to decay.

We restrict to the simplest class of extremal, supersymmetric black holes, namely those solutions with constant scalar fields. These solutions are known as *double extremal black holes*. In particular, as an indication of possible decay we study the ratio of masses between the non-supersymmetric double extremal black holes and their constituents.

To avoid non-constant scalar fields for the constituents, we mostly restrict to decay channels with BPS and anti-BPS objects.[2] We are able to demonstrate that this subclass of constituents provides viable decay channels for large families of extremal black holes. In addition, we also explore $R^2$ corrections to these decay channels.

More specifically, we consider compactifications of IIA string theory, with black holes formed as bound state of D$p$-branes supported on $p$-dimensional cycles of the Calabi-Yau threefold $X$. Black holes with vanishing D6-brane charge are amenable

---

[1]See for a comprehensive review, for example [16].

[2]Decay channels **4.2a** and **4.3e** include constituents which are extremal but neither BPS nor anti-BPS.

to analytic analysis, for example of the attractor points. We restrict to such black holes in this paper. The charge lattice contains supersymmetric cones, which contain the charges of supersymmetric black holes. The magnetic charges are carried by D4-branes and are positive for supersymmetric black holes. These correspond to holomorphic, effective divisors of $X$. For both supersymmetric [20, 21] and non-supersymmetric black holes [22] with positive D4-brane charges, the microscopic entropy is rather well understood in terms of the Maldacena-Strominger-Witten (MSW) conformal field theory (CFT). Such a description is not available for generic non-supersymmetric black holes in IIA supergravity.

By analyzing threshold masses in supergravity, we explore the stability of non-supersymmetric black holes in two families:

1. Black holes with positive magnetic charges, but electric D0-brane charge opposite to that of supersymmetric black holes. The charges of non-supersymmetric states of the MSW CFT lie in this cone of the charge lattice [23, 24]. Based on mass ratios, we demonstrate in Section 4.2, decay channel **4.2c**, that it is energetically favorable for such black holes to decay to a bound state of D0-branes and "polar" D0-D4-branes. The latter are themselves formed from bound states [25, 26]. We expect that the quantum-mechanical process for this decay is Schwinger pair creation of D0 and anti-D0-branes in the electric-magnetic field of the extremal black holes. Such pair creation in the background of Reissner-Nordström black holes has been discussed by [27, 28].

   Curiously, we find that including F-term $R^2$ corrections appears to make these decay channels less favorable. While this could be viewed as weakening the evidence for the WGC, we expect that a full analysis of $R^2$ corrections, including D-terms, is likely to be in better agreement with the predictions of the WGC.

2. Black holes with positive and negative magnetic charges. These charges correspond to non-holomorphic divisors. Analogously to five dimensions [29, 30], we find that the inequalities for decay depend crucially on geometric data of the compactification geometry, in particular the triple intersection numbers[3] and the positive cone of divisors. For $b_2(X) = 2$ and properly identifying the full effective cone of the CY threefold, we establish various valid decay channels at tree level, channels **4.4b** in Section 4.4.

While we find decay channels for various double extremal solutions, we are unable to identify such decay channels for a few specific cases of magnetically charged non-supersymmetric states. We expect that these are stable against decay, and give rise to recombination of (anti-)supersymmetric constituents to a non-supersymmetric

---

[3]These numbers are the entries of the triple intersection tensor $C_{abc}$ which determines the cubic prepotential (2.3) in supergravity.

bound state, as discussed recently also in five dimensions [29–31]. The interpretation in terms of the WGC is that these states are required to be purely quantum, or microscopic, ie they are elementary particles of the UV completion of the low-energy effective theory.

The paper is organised as follows. In Section 2 we briefly review the background material on $\mathcal{N} = 2$ supergravity and black holes in four dimensions. Section 3 is devoted to finding the attractor solutions for various Calabi-Yau compactifications. We then discuss various possible decay channels for these solutions in Section 4. The connection to the five-dimensional results is made in Section 5 and we conclude with a brief discussion and outlook in Section 6. In the Appendix we collect some useful formulas and computations.

## 2 Extremality and attractors

In this Section we review the physical setup we will be working in, namely black hole solutions in four-dimensional $\mathcal{N} = 2$ supergravity.

### 2.1 Review of $\mathcal{N} = 2$ supergravity in four dimensions

We consider type IIA string theory on a compact Calabi-Yau threefold (CY3), $X$, or, equivalently, M-theory on a circle times a CY3. This gives rise to $\mathcal{N} = 2$ supergravity in four dimensions with $h^{1,1}(X)$ vector multiplets, with $h^{i,j}(X)$ the Hodge numbers of $X$. The bosonic part of the supergravity action takes the form

$$
\begin{aligned}
S = \frac{1}{\kappa_4} \int_{\mathbb{R}^{1,3}} d^4x \sqrt{-G} \Big( & R - 2g_{a\bar{b}}(\partial t^a)(\partial \bar{t}^{\bar{b}}) \\
& - f_{AB}(t) F^A_{\mu\nu} F^{B\mu\nu} - \frac{1}{2}\tilde{f}_{AB}(t) F^A_{\mu\nu} F^B_{\rho\sigma} \epsilon^{\mu\nu\rho\sigma} \Big),
\end{aligned}
\tag{2.1}
$$

where $\kappa_4$ is the four-dimensional Newton's constant, $G$ the determinant of the spacetime metric, and $R$ the Riemann curvature. Moreover, $A, B = 0, \ldots, h^{1,1}(X)$, and $f_{AB}$, $\tilde{f}_{AB}$ are determined in terms of the prepotential $F$ introduced below [16, 17]. The metric on the complex moduli space $\mathcal{M}$ of Kähler moduli is $g_{a\bar{b}}$.

The complexified Kähler moduli $t^a$ are parametrized by the projective coordinates $X^A$,

$$
t^a := \frac{X^a}{X^0} = B^a + iJ^a,
\tag{2.2}
$$

where $B^a$ are the B-fields and $J^a$ are the (real) Kähler moduli, such that the Kähler form $J = J^a \omega_a$ with $\omega_a \in H^{1,1}(X)$ a basis of $H^{1,1}(X)$. The triple intersection numbers of the divisor dual to $\omega_a$ we denote by $C_{abc}$.

Let $F(X^A)$ be the prepotential of the theory. This function is homogenous of degree 2, $F(\lambda X^A) = \lambda^2 F(X^A)$ for $\lambda \in \mathbb{C}^*$. Using this symmetry, we can consider the gauge $X^0 = 1$. We will mostly be interested in the large volume limit, $J^a \to \infty$, where

we can neglect higher loop and instanton corrections. The perturbative prepotential is given by[4]

$$F(X^A) = \frac{1}{6} C_{abc} \frac{X^a X^b X^c}{X^0} + \frac{1}{24} \frac{1}{64} c_{2,a} \frac{X^a}{X^0} \hat{A},$$ (2.3)

where $c_{2,a}$ is the second Chern class of $X$, and $\hat{A}$ is a chiral background field related to the Weyl multiplet. The latter term involving $\hat{A}$ will give rise to a curvature squared (or $R^2$) correction in the effective action of supergravity. In most of our discussion, we reduce to tree level and set $c_{2,a} = 0$. In the gauge $X^0 = 1$, we have then

$$F(t^a) = \frac{1}{6} C_{abc} t^a t^b t^c.$$ (2.4)

The Kähler potential reads

$$K(X^A, \bar{X}^A) = -\log \left[ -i \left( (X^A)^* \partial_A F - X^A (\partial_A F)^* \right) \right].$$ (2.5)

At tree level, this evaluates to

$$K(X^A, \bar{X}^A) = -\log \left[ i \frac{1}{6} C_{abc} (t^a - \bar{t}^a)(t^b - \bar{t}^b)(t^c - \bar{t}^c) \right] = -\log \left[ 8 V_{IIA} \right],$$ (2.6)

with $V_{IIA}$ the tree level CY volume,

$$V_{IIA} = \frac{1}{6} C_{abc} J^a J^b J^c.$$ (2.7)

This volume is in string units and varies as function of the vector multiplet moduli [26]. For use in Sec. 5, we note that the volume in 11D Planck units belongs to a hypermultiplet, and is independent of the vector multiplet moduli. As a result, the volume is fixed in five-dimensional supergravity. For more details, see also Sec. 5.

The electric-magnetic charge of a $(D0, D2, D4, D6)$ brane bound state is denoted by

$$\gamma = (q_0, q_a, p^a, p^0) \in \mathbb{Q}^{2b_2 + 2}.$$ (2.8)

Sometimes it is also useful to consider $\gamma$ as a cohomology class,

$$\gamma = p^0 \omega_0 + p^a \omega_a + q_a \omega^a + q_0 \omega^0 \in \oplus_{j=0}^3 H^{2j}(X, \mathbb{Q}),$$ (2.9)

where $\omega_0$ is the generator of $H^0(X, \mathbb{Z})$, $\omega_a$ is a basis for $H^2(X, \mathbb{Z})$, $\omega^a$ a basis for $H^4(X, \mathbb{Z})$ and $\omega_0$ the generator for $H^6(X, \mathbb{Z})$. We will sometimes also consider the $(\omega_A, \omega^A)$ as basis elements for the Poincaré dual homology.

The superpotential is defined by

$$W(\gamma, X^A) = q_A X^A - p^A \partial_A F,$$ (2.10)

---

[4]Our convention for the prepotential follows the literature on extremal black holes [17, 18], and differs by a sign from some other literature, for example [16].

while the metric on the complexified Kähler moduli space, $\mathcal{M}$, is defined in terms of the Kähler potential as

$$g_{a\bar{b}} := \partial_a \partial_{\bar{b}} K. \tag{2.11}$$

We denote the central charge by $Z(\gamma, X^A, \bar{X}^A)$, defined as

$$Z(\gamma, X^A, \bar{X}^A) = e^{K/2} W(\gamma, X^A). \tag{2.12}$$

Upon a rescaling $X^A \to \lambda X^A$ with $\lambda \in \mathbb{C}^*$, we have $Z \to \lambda/|\lambda|\, Z$.

We note that $W$ is a holomorphic function (2.10) of the moduli $t$ (2.2) in the gauge $X^0 = 1$, and we will also use $W = W(\gamma, t)$. Similarly, we also use the notation $Z(\gamma, t, \bar{t}) = Z(\gamma, t)$ for $Z(\gamma, X^A, \bar{X}^A)$ and elsewhere, omitting the dependence on anti-holomorphic variables where appropriate.

The Kähler covariant derivative $\nabla_a Z$ of the central charge reads [18]

$$\nabla_a Z = \partial_a Z + \tfrac{1}{2}(\partial_a K)Z. \tag{2.13}$$

We have a simple relation between the covariant derivative of $Z$ and the derivative of $|Z|$,

$$
\begin{aligned}
\partial_a |Z| =& \bar{W}^{1/2}\left(\frac{1}{2}\frac{\partial_a W}{W^{1/2}} + \frac{1}{2}(\partial_a K)W^{1/2}\right)e^{K/2} = \frac{1}{2}e^{-i\alpha}\nabla_a Z, \\
\bar{\partial}_{\bar{a}}|Z| =& \frac{1}{2}e^{i\alpha}\bar{\nabla}_{\bar{a}}\bar{Z},
\end{aligned}
\tag{2.14}
$$

where $\alpha$ is the phase of $Z$ [32]

$$e^{i\alpha} = \frac{Z}{|Z|}.$$

Moreover, the covariant derivative acting on $W$ reads

$$\nabla_A W = \partial_A W + (\partial_A K)W. \tag{2.15}$$

## 2.2 Black hole solutions

We consider the static spherically symmetric metric [33]

$$ds^2 = e^{2U}dt^2 - e^{-2U}\left[\frac{c^4}{\sinh^4 c\tau}d\tau^2 + \frac{c^2}{\sinh^2 c\tau}d\Omega^2\right], \tag{2.16}$$

with $\tau \in (0, \infty)$ a parametrization of the radial direction, with $\tau \to 0$ at asymptotic infinity and $\tau \to \infty$ near the horizon.

A one-dimensional Lagrangian describing the radial evolution of $U$, $t$ and $\bar{t}$ as functions of $\tau$ can be derived from the two-derivative supergravity action. It is given by [17, 18]

$$\mathcal{L}(U, t^a, \bar{t}^{\bar{a}}) = \left(\frac{\partial U}{\partial \tau}\right)^2 + g_{a\bar{a}}\frac{\partial t^a}{\partial \tau}\frac{\partial \bar{t}^{\bar{a}}}{\partial \tau} + e^{2U}V_{BH}(\gamma, t). \tag{2.17}$$

The black hole potential $V_{BH}(\gamma, t)$ is a function of the charges and couplings of the theory and in the case of $\mathcal{N} = 2$ supergravity it takes the form

$$V_{BH}(\gamma, t) = g^{a\bar{b}} \nabla_a Z \,\bar{\nabla}_{\bar{b}} \bar{Z} + |Z|^2 = e^K \left[ g^{a\bar{b}} \nabla_a W (\nabla_b W)^* + |W|^2 \right], \qquad (2.18)$$

where $Z$ is the central charge (2.12).

The Lagrangian is supplemented by the constraint

$$\left( \frac{\partial U}{\partial \tau} \right)^2 + g_{a\bar{a}} \frac{\partial t^a}{\partial \tau} \frac{\partial \bar{t}^{\bar{a}}}{\partial \tau} - e^{2U} V_{BH}(\gamma, t) = c^2, \qquad (2.19)$$

where $c = 2ST$, with $S$ the entropy and $T$ the temperature of the black hole [34]. This condition is a manifestation of the first law of black hole thermodynamics, stating that the total energy of the system should be conserved.

The equations of motion from the Lagrangian (2.17) for $U$ and $t$ read

$$\partial_\tau^2 U = e^{2U} V_{BH},$$
$$e^{2U} \frac{\partial V_{BH}}{\partial \bar{t}^{\bar{b}}} = g_{a\bar{b}} \frac{\partial^2 t^a}{\partial \tau^2} + \left( \frac{\partial g_{a\bar{b}}}{\partial \bar{t}^{\bar{a}}} - \frac{\partial g_{a\bar{a}}}{\partial \bar{t}^{\bar{b}}} \right) \frac{\partial t^a}{\partial \tau} \frac{\partial \bar{t}^{\bar{a}}}{\partial \bar{\tau}} + \frac{\partial g_{a\bar{b}}}{\partial t^b} \frac{\partial t^a}{\partial \tau} \frac{\partial t^b}{\partial \tau}. \qquad (2.20)$$

When the moduli space is complex Kähler we have $\Gamma^a_{\bar{b}\bar{c}} = 0$ and the second equation simplifies. The Christoffel symbol of the Kähler metric is

$$\Gamma^a_{bc} = g^{a\bar{d}} \partial_b g_{c\bar{d}}, \qquad (2.21)$$

and we can write

$$\frac{\partial^2 t^a}{\partial \tau^2} + \Gamma^a_{bc} \frac{\partial t^b}{\partial \tau} \frac{\partial t^c}{\partial \tau} = g^{a\bar{b}} e^{2U} \frac{\partial V_{BH}}{\partial \bar{t}^{\bar{b}}}. \qquad (2.22)$$

The equations of motion are second order non-linear differential equations. Thus the initial conditions for $\tau = 0$ require the initial values as well as the initial first derivatives (or velocities).

The Lagrangian can alternatively be written as [33]

$$\mathcal{L} = \left( \partial_\tau U \pm e^U |Z| \right)^2 + \left| \partial_\tau t^a \pm e^{i\alpha} e^U g^{a\bar{b}} \bar{\nabla}_{\bar{j}} \bar{Z} \right|^2 \mp 2 \frac{d}{d\tau} \left( e^U |Z| \right), \qquad (2.23)$$

From this it is evident that the first order conditions

$$\partial_\tau U = -e^U |Z|,$$
$$\partial_\tau t^a = -e^{U+i\alpha} g^{a\bar{b}} \bar{\nabla}_{\bar{b}} \bar{Z}, \qquad (2.24)$$

minimize the Lagrangian. Here we fixed the sign, by requiring that $e^{-2U} \to \infty$ for $\tau \to \infty$. The constraint (2.19) vanishes, $c = 0$, if these linear equations are satisfied. These conditions are however not necessary for $c = 0$.

The system described above has a natural interpretation in classical mechanics as a particle moving in a $(2b_2 + 1)$-dimensional space, and the constraint (2.19) is the

conservation of kinetic plus potential energy. Here the potential energy is identified with $-e^{2U}V_{BH}$. Thus stable extrema correspond to maxima of $V_{BH}$, while minima of $V_{BH}$ are unstable. Since $\mathcal{L}$ does not contain a dissipative term, converging attractor solutions only occur if the total energy equals the maxima of $-e^{2U}V_{BH}$ such that the particle approaches the unstable maximum for $\tau \to 0$. This is in the BPS case ensured by the linear BPS equations (2.24), while for the non-BPS second order equations (2.22), such converging attractor behavior only occurs for the right choice of "initial velocity" $dt^a/d\tau|_{\tau=0}$.

We recall the following terminology:

- A black hole solution is *extremal*, if the constraint (2.19) is satisfied with $c = 0$.

- A black hole solution is *BPS*, if it satisfies the linear equation (2.24). In supergravity, such solutions are supersymmetric. These solutions are a subset of the extremal black holes.

- An extremal black hole solution is *double extremal*, if the scalar fields are independent of $r = 1/\tau$. Then $c^2/\sinh(c\tau)^2 \to 1/\tau^2$ in the metric (2.16) and $U(\tau) = -\log(1 + \sqrt{V_{BH}}\,\tau)$.

  The double extremal solutions are a subset of the extremal black holes. They come in two types

  1. *Double extremal BPS black holes*: BPS solutions, satisfying (2.24), for which $\partial t^a/\partial \tau = 0$, and thus $\nabla_a Z = 0$, throughout space-time.
  2. *Double extremal non-BPS black holes*: Solutions for which $\partial t^a/\partial \tau = 0$ throughout space-time, but not satisfying (2.24). As a result, $\bar{\partial}_{\bar{a}} V_{BH} = 0$ but $\nabla_a Z \neq 0$.

At spatial infinity, we have $\tau \to 0$, and $U \to M\tau$, with $M$ the ADM mass determined at asymptotic infinity. The metric becomes Minkowski for $r \to \infty$ and the constraint reads

$$
\begin{aligned}
&M(\gamma, t_\infty, \Sigma)^2 - |Z(\gamma, t_\infty)|^2 \\
&= c^2 + |\nabla_a Z(\gamma, t_\infty)|^2 - g_{a\bar{a}}\Sigma^a \bar{\Sigma}^{\bar{a}},
\end{aligned}
\tag{2.25}
$$

or

$$
M(\gamma, t_\infty, \Sigma)^2 = c^2 + V_{BH}(\gamma, t_\infty) - g_{a\bar{a}}\Sigma^a \bar{\Sigma}^{\bar{a}},
\tag{2.26}
$$

where we defined the scalar charge $\Sigma^a := \frac{dt^a}{d\tau}\Big|_{\tau=0}$ [18].

As mentioned above, extremal black holes have zero temperature and thus $c = 0$. The subset of BPS solutions satisfy the linear equations (2.24), and in particular

$$
\Sigma^a = -g^{a\bar{a}}\bar{\nabla}_{\bar{a}}\bar{Z}(\gamma, t_\infty).
\tag{2.27}
$$

So we reproduce the well-known relation between the mass and the central charge,

$$M(\gamma, t_\infty) = |Z(\gamma, t_\infty)|. \tag{2.28}$$

For non-BPS extremal black holes we still have $c = 0$, but $\Sigma^a \neq -g^{a\bar{a}}\bar{\nabla}_{\bar{a}}\bar{Z}(\gamma, t_\infty)$, and therefore also $M^2 \neq |Z|^2$. Eq. (2.26) gives an upperbound for the mass of these black holes, $M^2 \leq V_{BH}(\gamma, t_\infty)$. For the case of double extremal black holes $\Sigma^a = 0$, such that we have $M^2 = V_{BH}$. In this paper, we will only be concerned with decay channels for such double extremal black holes, either BPS or non-BPS.

## 2.3 Attractor equations

The determination of the attractor values of the moduli at the horizon, $\lim_{\tau \to \infty} t^a(\tau)$, is an important problem since these are necessary for the evaluation of the mass and entropy of double extremal black holes. Already for the linear BPS equations (2.24), this is in general a hard and non-trivial question [35] with interesting links to arithmetic geometry [36–38]. Recently, techniques have also been developed to include non-perturbative genus 0 instanton contributions [39]. In the non-BPS case, Eq. (2.20) demonstrates that the values at the horizon $t^a(\infty)$ minimize the effective potential [17, 18, 40]. Therefore, in order to find the attractor solutions we are interested in solving the equations

$$\partial_a V_{BH}(\gamma, t_\gamma) = e^K \left( g^{b\bar{c}}(\nabla_a \nabla_b W)\bar{\nabla}_{\bar{c}}\bar{W} + 2(\nabla_a W)\bar{W} \right) = 0. \tag{2.29}$$

The BPS attractors minimise the central charge such that $\nabla_A W = 0$, while the non-BPS attractors are the solutions to the above equations with $\nabla_A W \neq 0$. For the BPS attractor equation, $\nabla_A W = 0$ (2.15), we use that

$$\partial_A K = i\, e^K \left( (X^B)^* \partial_A \partial_B F - (\partial_A F)^* \right).$$

Then taking the real and imaginary part of $\nabla_A W = 0$ gives the well-known equations

$$q_A = 2\mathrm{Im}\!\left(e^{K/2}\,\bar{Z}\,F_A\right), \qquad p^A = 2\mathrm{Im}\!\left(e^{K/2}\,\bar{Z}\,X^A\right). \tag{2.30}$$

An important quantity when studying the attractor solutions is the matrix

$$m_{kl} = \frac{1}{2}\partial_k \partial_l V_{BH}(\gamma, t_\gamma), \tag{2.31}$$

and its eigenvalues, which are referred to as the mass matrix and masses of the scalar fields, respectively, in [17]. The indices $k, l$ run over the $2h^{1,1}(X)$ real dimensions of the Kähler moduli space. Here $t_\gamma$ refers to the critical values of the scalars. An attractor solution is a good solution if the eigenvalues of $m_{kl}$ are all positive.

## 2.4 D0-D2-D4 black holes

We will focus on D0-D2-D4 systems, and thus set the D6 brane charge, $p^0$, to zero. The microscopics of these black holes is described by the MSW CFT [20], and partition functions can be studied in detail [25, 41, 42, 44, 45]. To establish the attractor equations for both BPS and non-BPS black holes of this type, we first specialize various quantities to the case $p^0 = 0$. In the gauge $X^0 = 1$, we have for the tree level superpotential [5]

$$W = q_0 + q_a t^a - \frac{1}{2} C_{abc} p^a t^b t^c. \tag{2.32}$$

We introduce various shorthand notations

$$
\begin{aligned}
C_{ab} &= C_{abc} p^c, & C^{ab} C_{bc} &= \delta^a{}_c, & C_a &= C_{abc} p^b p^c, & C &= C_{abc} p^a p^b p^c, \\
L_{ab} &= C_{abc} J^c, & L^{ab} L_{bc} &= \delta^a{}_c, & L_a &= C_{abc} J^b J^c, & L &= C_{abc} J^a J^b J^c,
\end{aligned}
\tag{2.33}
$$

as well as the shifted variables

$$\hat{q}_0 := q_0 + \frac{1}{2} C^{ab} q_a q_b, \qquad \hat{t}^a := t^a - C^{ab} q_b, \tag{2.34}$$

or, since $C^{ab} q_b$ is real, $\hat{B}^a = B^a - C^{ab} q_b$. These shifts are motivated by a fractional spectral flow giving an effectively pure D0-D4 system [41, 42]. Finally, since $C_{ab}$ induces a quadratic form, of signature $(1, b_2 - 1)$, for elements $k_1, k_2 \in H^4(X, \mathbb{R})$, we will make use of the notation $C_{ab} k_1^a k_2^b = k_1 \cdot k_2$, and similar.

In Appendix A we give a few useful explicit formulas for the central charge and the black hole effective potential in terms of the charges and moduli for the D0-D2-D4 system. Using these formulas, we find the BPS condition

$$\nabla_a W = \frac{i}{4} \frac{L_a}{V_{IIA}} \left[ \hat{q}_0 - \frac{1}{2} \left( (\hat{B} \cdot \hat{B}) + 2i(J \cdot \hat{B}) - (J \cdot J) \right) \right] - C_{ab}(\hat{B}^b + iJ^b) = 0. \tag{2.35}$$

This is one set of equations for the real part and one set for the imaginary part. The real part tells us that (since $J^a > 0$ in the Kähler cone)

$$\frac{L_a}{4V_{IIA}} J \cdot \hat{B} = C_{ab} \hat{B}^b \implies \hat{B}^a = 0, \tag{2.36}$$

and the imaginary part gives the equation,

$$J^a(\hat{q}_0 + \frac{1}{2}(J \cdot J)) = 4 \, V_{IIA} \, p^a, \tag{2.37}$$

where we used $L^{ab} C_{bc} J^c = p^a$.

---

[5] In other places in the literature the holomorphic central charge is taken as $- \int_X e^{-t} \wedge \gamma$. This convention results in a different sign for $q_0$.

Extremising the full potential instead gives the condition that

$$
\begin{aligned}
\partial_a V_{BH} = {} & \frac{i}{4}\frac{L_a}{V_{IIA}}\left[(J\cdot J)^2 + (\hat{B}\cdot\hat{B})^2 + 2(J\cdot\hat{B})^2 + 4\hat{q}_0^2 - 4\hat{q}_0(\hat{B}\cdot\hat{B})\right] \\
& - 2iV_{IIA}\left[C_{afg}L^{bf}L^{cg}C_{bd}C_{ce}\hat{B}^d\hat{B}^e - 2iC_{ab}C_{cd}L^{bc}\hat{B}^d - C_{ab}p^b\right] \\
& + C_{ab}\left[2(J\cdot\hat{B})J^b + 2(\hat{B}\cdot\hat{B})\hat{B}^b - 2i(J\cdot J)J^b - 2i(J\cdot\hat{B})\hat{B}^b - 4\hat{q}_0\hat{B}^b\right].
\end{aligned}
\tag{2.38}
$$

vanishes. The real part now tells us that

$$
2V_{IIA}L^{ab}\hat{B}^c C_{bc} = J^a(J\cdot\hat{B}) + \hat{B}^a(\hat{B}\cdot\hat{B}) - 2\hat{q}_0\hat{B}^a,
\tag{2.39}
$$

with one solution being $\hat{B}^a = 0$. The imaginary part gives the condition

$$
\begin{aligned}
& \frac{L_a}{8V_{IIA}}\left[(J\cdot J)^2 + (\hat{B}\cdot\hat{B})^2 + 2(J\cdot\hat{B})^2 + 4\hat{q}_0^2 - 4\hat{q}_0(\hat{B}\cdot\hat{B})\right] \\
& = V_{IIA}C_{afg}L^{bf}L^{cg}C_{bd}C_{ce}\hat{B}^d\hat{B}^e - V_{IIA}C_{ab}p^b + C_{ab}J^b(J\cdot J) + C_{ab}\hat{B}^b(J\cdot\hat{B})
\end{aligned}
\tag{2.40}
$$

When $\hat{B}^a = 0$ this becomes

$$
J^a(4\hat{q}_0^2 + (J\cdot J)^2) = 8V_{IIA}\left((J\cdot J)p^a - V_{IIA}L^{ab}C_{bc}p^c\right).
\tag{2.41}
$$

In this paper, we focus our attention on this case and refer, in the following, to this equation as the *attractor equation*. For the one-moduli case, we show below that the solutions with $\hat{B}\neq 0$ are not viable.

This CFT is chiral and has $(0,4)$ supersymmetry. The entropy (4.8) follows from the Cardy formula with central charges $c_L = C + c_2\cdot p$ and $c_R = C + c_2\cdot p/2$. For a unitary CFT, the supersymmetric representations need to satisfy

$$
L_0 - \frac{c_L}{24} \geq 0, \qquad \bar{L}_0 - \frac{c_R}{24} \geq 0,
\tag{2.42}
$$

with $L_0, \bar{L}_0$ being the Virasoro generators. The momentum along the M-theory circle, or the D0 brane charge, is given as the difference between the Virasoro generators,

$$
q_0 = L_0 - \bar{L}_0 - \frac{c_L - c_R}{24}.
\tag{2.43}
$$

Putting this together we get a lower bound for $\hat{q}_0$ for the supersymmetric states,

$$
\hat{q}_0 \geq -\frac{c_L}{24}.
\tag{2.44}
$$

The Cardy formula gives the microscopic entropy for large $|\hat{q}_0|$ [20, 22–24],

$$
\begin{aligned}
\text{BPS:} \qquad & S_{CFT} = 2\pi\sqrt{\hat{q}_0\, c_L/6}, \\
\text{non-BPS:} \qquad & S_{CFT} = 2\pi\sqrt{-\hat{q}_0\, c_R/6}.
\end{aligned}
\tag{2.45}
$$

The equation for the BPS entropy obviously breaks down for $-c_L/24 \leq \hat{q}_0 < 0$. These states are supersymmetric, but do not consist of single black hole centers. Instead, they are bound states of multiple constituents. For example, the states with $\hat{q}_0 = -c_L/24$ are bound states of a D6-brane with an anti-D6 brane, with the D4-brane charge generated by a flux on the 6-brane worldvolume [25, 26]. Since all states corresponding to $\hat{q}_0 < 0$ are bound states of multiple constituents there is no BPS attractor point associated to such a total charge. On the other hand, the CFT states consist of those states at the large volume attractor point $t_\gamma^*$ [26],

$$(t_\gamma^*)^a = C^{ab} q_b + i p^a \lambda, \tag{2.46}$$

with $\lambda$ sufficiently large. We refer to the states with $-c_L/24 \leq \hat{q}_0 < 0$ as "polar D0-D4 states", since these states give rise to the so-called polar term in the partition function [41, 43].

## 3 Attractor solutions for one- and two-parameter models

In this section we study the attractor solutions for different families of Calabi-Yau manifolds.

### 3.1 The general class of attractor solutions

There is a general way to solve the attractor equations for any Calabi-Yau threefold [17, 35]. We will start by studying this solution. However, as we will see later this does not give all the non-supersymmetric solutions for the Calabi-Yau manifolds with $h^{1,1}(X) > 1$. The procedure is to first make the ansatz that $\hat{t}^a = i p^a z$, for some real parameter $z$ [17]. For this ansatz, the attractor equation, (2.41), reduces to

$$(\hat{q}_0 - \tfrac{1}{6} z^2 C)(\hat{q}_0 + \tfrac{1}{6} z^2 C) = 0. \tag{3.1}$$

The first factor corresponds to the BPS solution (satisfying $\nabla_a W = 0$) and the second to the non-BPS one (with $\nabla_a W \neq 0$). We thus have the two solutions [35][6]

$$\begin{aligned} \text{BPS:} \qquad & \hat{t}_\gamma^a = i p^a \sqrt{\frac{6\hat{q}_0}{C}}, \\ \text{non-BPS:} \qquad & \hat{t}_\gamma^a = i p^a \sqrt{-\frac{6\hat{q}_0}{C}}. \end{aligned} \tag{3.2}$$

We will refer to these as the "general" solutions in the following, since they hold for any Calabi-Yau. The Kähler cone condition tells us that $J^a = \operatorname{Im} t^a > 0$, which

---

[6]Here we choose the sign of the solution by requiring that $J^a$ should be in the Kähler cone, and thus positive. For the square root and other fractional powers, we will use the convention that the image of a positive real number is a positive real number.

thus means that we need $p^a > 0$ and $\pm \frac{6\hat{q}_0}{C} > 0$ for the BPS and non-BPS solutions, respectively. The non-BPS solutions have the same charges as the non-BPS states of the MSW CFT [20]. Therefore modulo decay of multi-center black holes in the decoupling limit [26], these solutions are captured by the MSW CFT.

It is possible to determine the effect of $R^2$ corrections to F-terms for the attractor values. For the BPS case, a closed expression is available [46], following Wald's formalism. For the non-BPS attractor values, an order by order analysis in $c_2$ can be carried out using the entropy formalism [47–49]. The results for both cases are

$$
\text{BPS:} \qquad \hat{t}^a_\gamma = ip^a \sqrt{\frac{6\hat{q}_0}{C + c_2 \cdot p}}, \qquad \hat{A}_\gamma = -\frac{64\, e^{-K(t_\gamma, \bar{t}_\gamma)}}{Z(\gamma, t_\gamma)^2},
$$

$$
\text{non-BPS:} \qquad \hat{t}^a_\gamma = ip^a \sqrt{-\frac{6\hat{q}_0}{C}\left(1 - \frac{9}{32}\frac{c_2 \cdot p}{C} + \dots \right)}, \qquad \hat{A}_\gamma = -\frac{4\, e^{-K(t_\gamma, \bar{t}_\gamma)}}{Z(\gamma, t_\gamma)^2}.
$$
$$(3.3)$$

Thus the magnitude of the Kähler modulus is reduced in both cases. It is important to note that we are only considering F-term corrections in the above, while non-supersymmetric black holes may also be affected by $R^2$ corrections to D-terms [50].

## 3.2 One-parameter CYs

Let us now turn to examples of Calabi-Yau threefolds with $h^{1,1}(X) = 1$. For ease of notation we define $\kappa := C_{111}$, $p := p^1$ and $q := q_1$.

For this simple case we can return to the generic equation for the minimising of the potentials. For the BPS case we saw in (2.37) that we need $\hat{B} = 0$, the BPS solution for $J$ is then

$$
J^2_\gamma = \frac{6\hat{q}_0}{\kappa p}.
$$
$$(3.4)$$

For the minimising of the full potential we saw that we can have either $\hat{B} = 0$ or $\hat{B} \neq 0$. The first case gives (2.41) and the solutions

$$
J^2_\gamma = \pm \frac{6\hat{q}_0}{\kappa p}.
$$
$$(3.5)$$

This reproduces the generic solution found in (3.2), with the minus sign again corresponding to the non-BPS solution. If we instead assume $\hat{B} \neq 0$ we find that (2.39) and (2.40) give

$$
\hat{B}^2_\gamma = \frac{18\hat{q}_0}{\kappa p},
$$
$$
J^2_\gamma = -\frac{24\hat{q}_0}{\kappa p}.
$$
$$(3.6)$$

However, this is not a viable attractor solution. One way to see this is that it gives opposite signs for $\hat{B}^2$ and $J^2$, even though both should be strictly positive for a viable

solution (i.e. $\hat{B}$ and $J$ should be real). It further gives negative eigenvalues for the mass matrix (2.31).

We thus conclude that for $h^{1,1}(X) = 1$, the only attractor solutions are the ones given by the general solution (3.2).

## 3.3 Two-parameter CICYs with autochthonous divisors

We now turn to considering the case of $h^{1,1}(X) = 2$. Since we now have two classes of divisors we can get new types of behaviour. It is in general hard to find all solutions to the full attractor equations (either BPS or non-BPS) for generic two-parameter CYs. To simplify, we consider only the cases with $\hat{B}^a = 0$, as in (2.41). Besides the generic solutions (3.2), we will see that there are further solutions once we lift the assumption that $\hat{t}^a \propto p^a$. If we look at classes of CYs where certain intersection numbers vanish, we can more easily solve the general equations without making this assumption. To this end, let us start looking at complete intersection Calabi-Yau (CICY) manifolds in $\mathbb{P}^1 \times \mathbb{P}^n$. As we discuss below, these CYs have $C_{111} = C_{112} = 0$ and the generators of their effective cone include an autochtonous divisor.[7] This class of solutions correspond to 16 of the CICYs of [51]. There are also 10 toric hypersurface Calabi-Yau (THCY) manifolds in [51] that will satisfy the same conditions, and thus have the same solutions for the attractor equations. See also Tables 2 and 3 of the Appendix. In this paper we focus the discussion on the CICYs, but the analysis for the THCY gives the same results.

The CICYs of the class considered here are all constructed in an ambient space of the type $\mathcal{A} = \mathbb{P}^1 \times \mathbb{P}^n$, for some $n > 1$. We thus have an embedding of the Calabi-Yau $X$ in $\mathcal{A}$, $f : X \to \mathcal{A}$. Assuming that the embedding satisfies the conditions of the Lefshetz hyperplane theorem, the cohomology of $X$, $H^r(X, \mathbb{Q})$ is isomorphic to $H^r(\mathcal{A}, \mathbb{Q})$ for $r \leq 2$. Thus in particular, the cohomology of 2-forms is isomorphic, and two generators $\omega_j$ are pull-backs from 2-forms on $\mathcal{A}$, $\omega_a = f^* \eta_a$. Thus, the map $f^* : H^2(\mathcal{A}, \mathbb{Q}) \to H^2(X, \mathbb{Q})$ combined with Poincaré duality gives a map $\tilde{f}^* : H_{2n}(\mathcal{A}, \mathbb{Q}) \to H_2(X, \mathbb{Q})$.

Our main interest is in the effective cone $C(X) \subset H_4(X, \mathbb{Z})$. The effective cone $C(\mathcal{A}) \subset H_{2n}(\mathcal{A}, \mathbb{Z})$ is spanned by two divisors $\mathcal{D}_1$ and $\mathcal{D}_2$. We take these to be $\mathcal{D}_1 \simeq \mathbb{P}^n \subset \mathcal{A}$ and $\mathcal{D}_2 \simeq \mathbb{P}^1 \times \mathcal{H}_{\mathbb{P}^n}$, with $\mathcal{H}_{\mathbb{P}^n} \in H_{2n-2}(\mathbb{P}^n)$ the hyperplane of $\mathbb{P}^n$. Using $\tilde{f}^*$ introduced above, we obtain effective divisors of $X$ by $D_a = \tilde{f}^* \mathcal{D}_a$ for $a = 1, 2$. Clearly, $D_1$ does not self-intersect, and therefore the intersection numbers $C_{111}$ and $C_{112}$ of $X$ vanish.

It is *not* true in general that $D_1$ and $D_2$ are the generators of the effective cone $C(X)$. Generically, the effective cone $C(X)$ is enlarged from that generated by the divisors $D_1$ and $D_2$ inherited from the ambient space. The effective cone $C(X)$ is

---

[7]Since the generic expressions such as (2.41) are of course symmetric in $J^1$ and $J^2$ we get the same type of solutions with the indices 1 and 2 interchanged.

instead generated by $D_1$ and an exceptional divisor, $D_3$, of the form $D_3 = mD_2 - D_1$ for some $m$ [51, 52]. See Appendix B for the explicit forms for the different Calabi-Yau manifolds we consider. This third divisor is sometimes referred to as *autochthonous*, since it is not inherited from an effective divisor on the ambient space [53]. Similarly for the THCYs of Table 3, the effective cones are also generated by a third exceptional divisor (together with one of the divisors inherited from the ambient space), of the same form. We will see later that this third divisor plays an important role when studying the possible decays of the non-supersymmetric black holes [29].

Among the solutions that minimise the effective potential are of course the general solutions (3.2). In addition, we find for this class of CYs the "particular solution",

$$
\begin{aligned}
J_\gamma^1 &= -\frac{1}{3C_{122}}(3C_{122}p^1 + 2C_{222}p^2)\sqrt{-\frac{6\hat{q}_0}{C}}, \\
J_\gamma^2 &= p^2\sqrt{-\frac{6\hat{q}_0}{C}}.
\end{aligned}
\tag{3.7}
$$

This solution can be obtained from the general BPS solution (3.2) by the map

$$
p^1 \mapsto -\frac{1}{3C_{122}}(3C_{122}p^1 + 2C_{222}p^2), \qquad p^2 \mapsto p^2.
\tag{3.8}
$$

This transformation keeps $|C|$ invariant but changes the sign of $C$, $C = 3C_{122}p^1(p^2)^2 + C_{222}(p^2)^3 \mapsto -C$. We will discuss in Section 5 how this solution is related to the 5-dimensional solutions of [29, 30].

Let us consider the domain of the charges for which (3.7) can be a proper solution. For the moduli to be in the Kähler cone, we need $p^2 > 0$ and $-\frac{1}{3C_{122}}(3C_{122}p^1 + 2C_{222}p^2) > 0$. Since $C_{abc} > 0$ for all $a, b, c$, except for $C_{111} = C_{112} = 0$ and permutations, then $-3C_{122}p^1 > 2C_{222}p^2 > 0$, such that $p^1$ and $p^2$ have opposite signs. As a result, there is no overlap between the charge domains for (3.7) and the general solution for which the $p^a$ have the same sign. For (3.7), we have furthermore that $C < 0$, such that for this solution to be in the Kähler cone we also need $\hat{q}_0 > 0$ (contrary to the general non-BPS solution). The three possible attractor solutions for this class of Calabi-Yau manifolds thus live in three separate charge sectors given by

$$
\begin{aligned}
\text{BPS:} \quad & p^a > 0, \ \hat{q}_0 > 0, \\
\text{General non-BPS:} \quad & p^a > 0, \ \hat{q}_0 < 0, \\
\text{Particular non-BPS:} \quad & p^2 > 0, \ -p^1 > \frac{2C_{222}}{3C_{122}}p^2 > 0, \ \hat{q}_0 > 0.
\end{aligned}
\tag{3.9}
$$

We note here that, if we instead have $C_{122} = C_{222} = 0$, we get the same results as above with $C_{222}$ interchanged with $C_{111}$, and $C_{122}$ with $C_{112}$. In particular, even though we focused the discussion around the complete intersection Calabi-Yaus of Table 2, the same results hold for all the toric hypersurfaces of Table 3.

### 3.4 Two-parameter families without autochthonous divisors

There are two THCY and one CICY in the classification of [51] that do not get an enlarged Kähler cone due to the presence of an autochthonous divisor, these all have $C_{111} = C_{222} = 0$. See Table 4. This means that the effective cones of these manifolds are generated simply by $D_1$ and $D_2$, and these generators have no self-intersection. In five dimensions these allow for stable non-BPS black strings according to the analysis of [29, 30]. As before, we have the general solutions (3.2). In addition there are particular solutions, which are of a different flavor than the class of CYs in Subsection 3.3. Finding all solutions of the attractor equations (2.41) for $J^a$ is complicated. To obtain solutions, we start by studying the solutions in terms of the charges $p^a$. For the case of $C_{111} = C_{222} = 0$, Eq. (2.41) gives

$$
\begin{aligned}
p^1 &= \frac{2\hat{q}_0(C_{112}J_1 + 2C_{122}J_2)(2C_{112}^2J_1^2 + C_{112}C_{122}J_1J_2 + C_{122}^2J_2^2)}{J_2(C_{112}J_1 + C_{122}J_2)\sqrt{H(J_1, J_2)}}, \\
p^2 &= -\frac{2\hat{q}_0(2C_{112}J_1 + C_{122}J_2)(C_{112}^2J_1^2 + C_{112}C_{122}J_1J_2 + 2C_{122}^2J_2^2)}{J_1(C_{112}J_1 + C_{122}J_2)\sqrt{H(J_1, J_2)}},
\end{aligned}
\tag{3.10}
$$

where we defined

$$
\begin{aligned}
H(J_1, J_2) &:= 4C_{112}^6J_1^6 + 12C_{112}^5C_{122}J_1^5J_2 + 21C_{112}^4C_{122}^2J_1^4J_2^2 \\
&\quad + 22C_{112}^3C_{122}^3J_1^3J_2^3 + 21C_{112}^2C_{122}^4 + 12C_{112}C_{122}^5J_1J_2^5 + 4C_{122}^6J_2^6,
\end{aligned}
\tag{3.11}
$$

for brevity. Note that, similar to the solutions we found for $C_{111} = C_{112} = 0$, we must have $\mathrm{sgn}(p^1) \neq \mathrm{sgn}(p^2)$ in order for this solution to be in the Kähler cone. As stated above, the inverted solutions in terms of $J^a$ are generally not tractable, but using the above solutions we can find a few special cases where things simplify. To illustrate this, we set $p^2 = -4p^1 < 0$ and consider the THCY with $C_{112} = C_{122} = 1$, referred to as $(1,1)_{-54}^{2,29}$ in [51]. Entry two of Table 4. For this particular set of charges the solutions for $J^a$ are easy to find,

$$
\begin{aligned}
J_\gamma^1 &= \left(\frac{239 - 57\sqrt{17}}{236}\right)^{1/4}\sqrt{\frac{\hat{q}_0}{p^1}}, \\
J_\gamma^2 &= \frac{3 + \sqrt{17}}{2}\left(\frac{239 - 57\sqrt{17}}{236}\right)^{1/4}\sqrt{\frac{\hat{q}_0}{p^1}}.
\end{aligned}
\tag{3.12}
$$

## 4 Double extremal black holes and decay channels

In this section, we consider decay channels for extremal black holes. An extremal black hole can decay if the sum of the masses of the decay products $\sum_j M_j$ is smaller or equal to the total mass $M$ of the black hole under consideration. Thus if the ratio

$$
\mathcal{R} = \frac{M^2}{(\sum_j M_j)^2},
\tag{4.1}
$$

is larger than 1 decay is energetically favorable. The simplest class of extremal black holes for which this ratio is readily determined are the double extremal black holes, whose mass squared $M^2$ is given by $V_{BH}$ (2.26) and the moduli are constant, given by the attractor values $t_\gamma$.

To avoid non-trivial attractor flows for the decay products, we consider BPS and anti-BPS objects as constituents. For such objects, the mass simplifies and is given by the absolute value of the central charge, which is also easily determined for non-constant attractor flows. We thus consider an extremal, non-BPS black hole with charge $\gamma$ as a bound state of BPS and anti-BPS states with charges $\gamma_j$, such that $\sum_j \gamma_j = \gamma$. For such decay channels, the ratio $\mathcal{R}$ (4.1) becomes

$$\mathcal{R}(\gamma, \{\gamma_j\}) := \frac{V_{BH}(\gamma, t_\gamma)}{(\sum_j |Z(\gamma_j, t_\gamma)|)^2}, \tag{4.2}$$

where $Z(\gamma_j, t_\gamma)$ (2.12) is the central charge of the (anti-)BPS state with charge $\gamma_j$ evaluated at the attractor point $t_\gamma$ for the total charge $\gamma$. If $\mathcal{R} > 1$, it is energetically favorable for the non-BPS state to decay into the (anti-)BPS constituents, while if $\mathcal{R} < 1$ the state is stable. The case $\mathcal{R} = 1$ could be considered as a threshold bound state, and we will find various threshold decay channels.

If all charges correspond to D-branes of the same dimension, the numerator of $\mathcal{R}$ (4.1) is mathematically the volume squared of a submanifold in $X$ with homology class $\gamma$, whose volume is at a local minimum as function of the moduli parametrizing the embedding of the submanifold. The denominator of $\mathcal{R}$ (4.2) for (anti-)BPS constituents is the volume squared of the "piece-wise calibrated cycle", that is to say the volume of the linear combination of holomorphic and anti-holomorphic cycles, whose homology class adds up to $\gamma$.

## 4.1 The general BPS solutions

We review and collect a few results for BPS solutions. At the BPS attractor point $t_\gamma^a$ (3.2), the exponentiated Kähler potential evaluates to

$$e^{K(t_\gamma, \bar{t}_\gamma)/2} = \frac{1}{\sqrt{8V_{IIA}}} = \frac{1}{2\sqrt{2}} \left(\frac{C}{6\hat{q}_0^3}\right)^{1/4}. \tag{4.3}$$

The holomorphic central charge reads

$$W(t_\gamma, \gamma) = 4\hat{q}_0. \tag{4.4}$$

For the central charge we thus find

$$Z(\gamma, t_\gamma) = (2\hat{q}_0 C/3)^{1/4}. \tag{4.5}$$

And the mass of the double extremal BPS black hole,

$$M_{BPS} = |Z(\gamma, t_\gamma)| = (2\hat{q}_0 C/3)^{1/4}. \tag{4.6}$$

This reproduces the well-known black hole entropy at tree level $S_{BH} = \pi\sqrt{2\hat{q}_0 C/3}$ (2.45) [20]. If we include the $R^2$ correction, the mass becomes

$$M_{BPS} = (2\hat{q}_0(C + c_2 \cdot p)/3)^{1/4}, \tag{4.7}$$

and the entropy

$$S_{BH} = \pi\sqrt{2\hat{q}_0(C + c_2 \cdot p)/3}. \tag{4.8}$$

Again in agreement with the microscopic entropy (2.45) [20, 46].

## 4.2 The general non-BPS solutions

This subsection considers decay channels for non-BPS states with generic attractor point (3.2). We will find that these black holes can decay at tree level to D0's and polar D0-D4 states.

We start by determining various quantitites. Using the general attractor moduli, $t^a_\gamma = ip^a\sqrt{-\frac{6\hat{q}_0}{C}} + C^{ab}q_b$ (3.2) of the non-BPS state we can evaluate the Kähler potential

$$e^{K(t_\gamma,\bar{t}_\gamma)/2} = \frac{1}{\sqrt{8V_{IIA}}} = \frac{1}{2\sqrt{2}}\left(-\frac{C}{6\hat{q}_0^3}\right)^{1/4}. \tag{4.9}$$

The central charge function for this moduli for some set of charges $\tilde{\gamma} = (\tilde{q}_0, \tilde{q}_a, \tilde{p}^a)$ is then

$$
\begin{aligned}
Z(\tilde{\gamma}, t_\gamma) = &\frac{1}{2\sqrt{2}}\left(-\frac{C}{6\hat{q}_0^3}\right)^{1/4}\left[\tilde{q}_0 + i\tilde{q}_a p^a\sqrt{-\frac{6\hat{q}_0}{C}} + \tilde{q}_a C^{ab}q_b\right.\\
&\left. - \frac{1}{2}\left(\tilde{p}^a p^b C_{ab}\frac{6\hat{q}_0}{C} + 2i\tilde{p}^a q_a\sqrt{-\frac{6\hat{q}_0}{C}} + C_{abc}\tilde{p}^a C^{bd}C^{ce}q_d q_e\right)\right].
\end{aligned}
\tag{4.10}
$$

If we set $\tilde{\gamma} = \gamma$ we get the central charge of the non-BPS black hole at the attractor point,

$$Z(\gamma, t_\gamma) = -\frac{1}{\sqrt{2}}(-\tfrac{1}{6}C\hat{q}_0)^{1/4}. \tag{4.11}$$

We can also calculate the effective potential by using the formulas given in Appendix A and evaluate them at the attractor point $t^a_\gamma$,

$$
\begin{aligned}
W(\gamma, t_\gamma) = &-2\hat{q}_0,\\
\nabla_a W(\gamma, t_\gamma) = &\frac{1}{2}iC_{ab}p^b\sqrt{-\frac{6\hat{q}_0}{C}},\\
g_{a\bar{b}}(t_\gamma, \bar{t}_\gamma) = &-\frac{1}{8\hat{q}_0 C}(2C_{ab}C - 3C_{ac}C_{bd}p^c p^d),\\
g^{a\bar{b}}(t_\gamma, \bar{t}_\gamma) = &-\frac{2}{3}\sqrt{-\frac{6\hat{q}_0^3}{C}}\left(6\sqrt{-\frac{C}{6\hat{q}_0}}C^{ab} - 3p^a p^b\sqrt{-\frac{6}{\hat{q}_0 C}}\right).
\end{aligned}
\tag{4.12}
$$

Note that the absolute value of the central charge $W$ is smaller for the non-BPS black hole than for the BPS black hole (4.4).

One verifies using these equations that the effective potential evaluates to

$$V_{BH}(\gamma, t_\gamma) = M^2_{non-BPS} = 16\,\hat{q}_0^2\,e^K = \sqrt{-2\hat{q}_0 C/3}. \tag{4.13}$$

The entropy $S_{BH} = \pi\sqrt{-2\hat{q}_0 C/3}$ agrees with the leading term of the microscopic entropy (2.45) [22–24].

We can compare this to the central charge squared, of the non-BPS object, (4.11), which gives

$$\frac{V_{BH}(\gamma, t_\gamma)}{|Z(\gamma, t_\gamma)|^2} = 4. \tag{4.14}$$

Thus the tree level mass of this non-BPS black hole is exactly twice the magnitude of its central charge.

**Channel 4.2a: Decay into non-BPS constituents**
When the $R^2$ corrections from the vector multiplet sector to the non-BPS mass are included, the mass squared is no longer simply related by $V_{BH}$ as in (2.26). The correction to the non-BPS mass is determined to first order in [54]. It reads

$$M_{non-BPS} = (-2\hat{q}_0 C/3)^{1/4}\left(1 - \frac{3}{320}\frac{c_2 \cdot p}{C} + \dots\right) \tag{4.15}$$

The negative sign does make such black holes unstable for decay into lighter non-BPS constituents as suggested by WGC, at least if the charge of the black hole $\gamma$ is parallel to that of the constituents $\gamma_j$. In that case, the non-BPS black hole as well as the non-BPS constituents are double extremal. For example, if we consider the decay $\gamma = n\gamma' \to n \times \gamma'$, we have for the ratio $\mathcal{R}$ (4.1),

$$\mathcal{R}(\gamma, \{n \times \gamma'\}) = 1 + (n^2 - 1)\frac{3}{160}\frac{c_2 \cdot p}{C} + \dots > 1. \tag{4.16}$$

Thus for increasing charge the ratio $\mathcal{R}$ decreases, and non-BPS black holes of this type can decay to non-BPS states with charge vectors whose entries are relatively prime.

**Channel 4.2b: Decay into D0's and D4's**
We proceed by considering decay of the non-BPS bound state into a number of D0-branes, and separate D4-branes (here we assume that the D2-brane charge vanishes). Using (4.10), we have for the central charges

$$Z_0 := Z(\gamma_0, t_\gamma) = -\frac{1}{2\sqrt{2}}\left(-\frac{Cq_0}{6}\right)^{1/4},$$

$$Z_a := Z(\gamma_a, t_\gamma) = \frac{1}{2\sqrt{2}}\left(-\frac{Cq_0}{6}\right)^{1/4}\left(\frac{3p^a C_a}{C}\right), \qquad \text{(no sum over } a\text{)}, \tag{4.17}$$

where the notation is that $Z_0$ is the central charge of a D0-brane with charge $\gamma_0 = (q_0, 0, 0)$, while $Z_a$ is the central charge function of a D4-brane wrapping the $a$th divisor, i.e. with $\gamma_a = (0, 0, \ldots, p^a, 0, \ldots)$. The mass ratio (4.2) with these constituents is then

$$\mathcal{R}(\gamma, \{\gamma_0, \gamma_a\}) = \frac{V_{BH}(\gamma, t_\gamma)}{\left(|Z_0| + \sum_{a=1}^{h^{1,1}} |Z_a|\right)^2} = \frac{16}{\left(1 + 3\sum_{a=1}^{h^{1,1}} \left|\frac{C_a p^a}{C}\right|\right)^2}, \tag{4.18}$$

where we again do not make use of the Einstein summation convention in the denominator. Since $C$ and $C_a p^a$ are both positive for all $a$, the sum over $a$ evaluates to 1, and we arrive at

$$\mathcal{R}(\gamma, \{\gamma_0, \gamma_a\}) = 1. \tag{4.19}$$

This indicates that the double extremal non-BPS magnetic black hole could be considered as a threshold bound state of D0 and D4 BPS constituents.

Similar results were analysed in great detail for the STU model in [55], where the only non-zero intersection number is $C_{123}$ and we indeed have that the above ratio is equal to unity. For the STU model the authors of [55] make use of its U-duality group to argue that the result for the D0-D4 system is generic.

One may wonder whether $R^2$ corrections alter the conclusion of threshold stability. Including the first order $R^2$ corrections in (4.17) using (3.3), one obtains:

$$\begin{aligned}
Z_0 &= -\frac{1}{2\sqrt{2}} \left(-\frac{Cq_0}{6}\right)^{1/4} \left(1 + \frac{23}{64} \frac{c_2 \cdot p}{C} + \ldots\right), \\
Z_a &= \frac{1}{2\sqrt{2}} \left(-\frac{Cq_0}{6}\right)^{1/4} \left(\frac{3p^a C_a}{C}\right) \left(1 - \frac{23}{192} \frac{c_2 \cdot p}{C} + \ldots\right),
\end{aligned} \tag{4.20}$$

with again no sum over $a$. Thus the effect of $R^2$ corrections is that the mass of the D0-brane increases, since the $V_{IIA}$ decreases, while the mass of the D4-brane decreases. Interestingly, if we sum up $|Z_0| + \sum_a |Z_a|$, the first order corrections cancel exactly. As a result, while $\mathcal{R} = 1$ (4.18) before including corrections, it becomes

$$\mathcal{R}(\gamma, \{\gamma_0, \gamma_p\}) = 1 - \frac{3}{160} \frac{c_2 \cdot p}{C} + \ldots \tag{4.21}$$

after including corrections. The $R^2$ corrections thus make decay in this orginally threshold channel less likely. This possibly suggests that different constituents, possibly non-(anti)-BPS, give viable decay channels instead of this one.

**Channel 4.2c: Decay into D0's and D0-D4's**
Since the $R^2$ corrections do not improve decay in the channels studied above, we explore other decay channels at tree level. In particular decay into a supersymmetric D0-D4 state with charge $\gamma_p = (\tilde{q}_0, 0, p^a)$, and a D0-branes with charge $\gamma_0 =$

$(q_0 - \tilde{q}_0, 0, 0)$. Then we have for $Z_0$ and $Z_p$,

$$Z_0 = \frac{1}{2\sqrt{2}} \left( -\frac{C}{6q_0^3} \right)^{1/4} (q_0 - \tilde{q}_0),$$

$$Z_p = \frac{1}{2\sqrt{2}} \left( -\frac{C}{6q_0^3} \right)^{1/4} (\tilde{q}_0 - 3q_0). \tag{4.22}$$

Now if we determine the ratio $R$ for this decay, we find

$$\mathcal{R}(\gamma, \{\gamma_0, \gamma_p\}) = \frac{V_{BH}(\gamma, t_\gamma)}{(|Z_0| + |Z_p|)^2} = \left( 1 - \frac{\tilde{q}_0}{2q_0} \right)^{-1} = 1 + \frac{\tilde{q}_0}{2q_0} + \dots, \tag{4.23}$$

where we assumed $|\tilde{q}_0/q_0| < 1$. Thus we see that if $\tilde{q}_0$ has the same sign as $q_0$, and thus negative, $\mathcal{R} > 1$, and these constituents give rise to a proper decay channel. Eq. (2.44) demonstrates that states with $\tilde{q}_0$ and $C > 0$ indeed exists, these are the "polar" D0-D4 states. As explained there, these states are not black hole solutions with a single black hole center, but instead bound states of multiple constituents, such as D6, anti-D6 and D0-branes.

If we include $R^2$ interactions, there will be a competition between the negative contribution of (4.21) and the positive contribution of (4.23), even though the $R^2$ correction differs from $\mathcal{R} = 1$ with $O(\text{charge}^{-2})$, and the tree level (4.23) differs from 1 with $O(\text{charge}^0)$. Since $q_0$ is unbounded below but $\tilde{q}_0$ is bounded below by $-C/24$, the term $\tilde{q}_0/q_0$ in (4.23) can be arbitrarily small. On the other hand, further $R^2$ corrections are expected beyond those of the vector multiplets considered here. So the results here are not conclusive.

We will study the effect of turning on the D2-brane charge for the decay of the one-parameter threefolds below. When we go to threefolds with $h^{1,1} > 1$ more possibilities will be available, as we discuss in Sec. 4.4.

## 4.3 One-parameter models and dyonic black holes

To reach exact expressions for $\mathcal{R}$, let us consider the one-parameter models in this section. For these threefolds, there are only the general solutions (3.5) to the attractor equations. For the double extremal case, the mass of the BPS black hole is given by (4.6) with $C = \kappa p^3$, while the mass of the non-BPS black holes is given by (4.13).

The central charge of the D0-D2-D4 system, for some charges $\tilde{\gamma} = (\tilde{q}_0, \tilde{q}, \tilde{p})$, at the non-BPS attractor moduli is, from (4.10),

$$Z(\tilde{\gamma}, t_\gamma) = e^{K/2} \left[ \tilde{q}_0 + i(\tilde{q}p - q\tilde{p})A + \frac{q\tilde{q}}{\kappa p} - \frac{q^2\tilde{p}}{2\kappa p^2} - 3\hat{q}_0 \frac{\tilde{p}}{p} \right] \tag{4.24}$$

with $A = \sqrt{-\frac{6\hat{q}_0}{\kappa p^3}}$.

**Channel 4.3a: Decay into D0-D4 and anti-D0-D4**

If we consider first the case of no D2 branes,[8] since there is only one divisor, the only possible (anti-)BPS bound state constituents are either that we have the D0 and D4 branes separate, giving the mass ratio (4.23), as we saw in the previous Section.

Alternatively, we can consider decay into a BPS state D0-D4 brane and anti-BPS state D0-D4-brane,

$$
\begin{aligned}
\gamma_1 &= (-xq_0, 0, zp), \\
\gamma_2 &= ((1+x)q_0, 0, (1-z)p).
\end{aligned}
\tag{4.25}
$$

for some $x \geq 0$ and $z \geq 1$. This gives the mass ratio

$$
\mathcal{R}(\gamma, \{\gamma_1, \gamma_2\}) = \frac{V_{BH}(\gamma, t_\gamma)}{(|Z(\gamma_1, t_\gamma)| + |Z(\gamma_2, t_\gamma)|)^2} = \frac{4}{(x + 3z - 1)^2} \leq 1,
\tag{4.26}
$$

with the saturation happening at $x = 0$, $z = 1$, which is the situation where the D0- and D4-branes are part of separate constituents, as before.

### Channel 4.3b: Decay into D0-, D2- and D4-branes

We proceed by letting the D2-brane charges be generic. Then we can, for example, consider constituents with charges

$$
\begin{aligned}
\gamma_0 &= (q_0, 0, 0), \\
\gamma_q &= (0, q, 0), \\
\gamma_p &= (0, 0, p).
\end{aligned}
\tag{4.27}
$$

The corresponding central charges are

$$
Z_0 \coloneqq Z(\gamma_0, t_\gamma), \qquad Z_q \coloneqq Z(\gamma_q, t_\gamma) \qquad Z_p \coloneqq Z(\gamma_p, t_\gamma),
\tag{4.28}
$$

we get the mass ratio

$$
\begin{aligned}
\mathcal{R}(\gamma, \{\gamma_0, \gamma_q, \gamma_p\}) &= \frac{V_{BH}(\gamma, t_\gamma)}{(|Z_0| + |Z_q| + |Z_p|)^2} \\
&= \frac{16(q_0 + \frac{q^2}{2kp})^2}{(|q_0| + \left|\frac{q^2}{kp} + iqpA\right| + \left|3q_0 + 2\frac{q^2}{kp} + iqpA\right|)^2}.
\end{aligned}
\tag{4.29}
$$

This is equal to one when $q = 0$ but smaller than one when $q \neq 0$, such that it does not correspond to an allowed decay channel for the dyonic black hole.

### Channel 4.3c: Decay into D0-D2-D4 and anti-D0-D2-D4

Alternatively, we can consider the decay products with charges

$$
\begin{aligned}
\gamma_1 &= (-xq_0, yq, zp), \\
\gamma_2 &= ((1+x)q_0, (1-y)q, (1-z)p),
\end{aligned}
\tag{4.30}
$$

---

[8]Equivalently we can consider the situation where we have some (anti-)D2 branes that form a bound state with the D0 branes. We then simply put hats on the relevant factors.

for some $x \geq 0$, $z \geq 1$ and $y \in \mathbb{Q}$. The central charges $Z_j$ are,

$$Z_1 := Z(\gamma_1, t_\gamma), \qquad Z_2 := Z(\gamma_2, t_\gamma). \tag{4.31}$$

This means that $\gamma_1$ is a BPS D0-D2-D4 state while $\gamma_2$ is an anti-BPS state. The mass ratio,

$$\mathcal{R}(\gamma, \{\gamma_1, \gamma_2\}) = \frac{V_{BH}(\gamma, t_\gamma)}{(|Z_1| + |Z_2|)^2}, \tag{4.32}$$

only has a maximum equal to one for certain values of $q$. For example, if $q_0 = -50000$, $k = 5$ and $p = 10$, the maximum is only equal to one if $q \in \{-1, 0, 1\}$, and this happens when $x = 0$, $y = \frac{3}{4}$ and $z = 1$. If $|q| > 1$ the maximum is smaller than one.

**Channel 4.3d: Decay into D0-D2-D4 and anti-D0-D2-D4**
Another possible situation we can consider is where the initial non-BPS state has no D2 charge but decays into two BPS states with D2 charges, of course adding up to zero. We thus consider two BPS states with charges $\gamma_1$ and $\gamma_2$ and $\gamma = \gamma_1 + \gamma_2$. The central charges are

$$Z_1 := Z(\gamma_1, t_\gamma), \qquad Z_2 := Z(\gamma_2, t_\gamma), \tag{4.33}$$

with

$$\begin{aligned}
\gamma_1 &= (-xq_0, yq, zp), \\
\gamma_2 &= ((1+x)q_0, -yq, (1-z)p),
\end{aligned} \tag{4.34}$$

for $x \geq 0$, $y \in \mathbb{Q}$ and $z \geq 1$. The mass ratio with these states however also does not reach a value larger than one. It exactly becomes one only when $x = y = 0$ and $z = 1$, which corresponds to the previous situation where we have no electric charge in the decay constituents and the D0- and D4-branes are split.

**Channel 4.3e: Decay into non-BPS D0-D4 and BPS D0-D4**
We briefly explore here the case of a decay channel for a D0-D4 double extremal black hole with charge $\gamma = (q_0, 0, p)$, with as constituents an extremal, but non-(anti)-BPS D0-D4 black hole with charge $\tilde{\gamma} = (\tilde{q}_0, 0, \tilde{p})$, and a BPS D0-D4 black hole with charge $\gamma - \tilde{\gamma}$. The ratio of interest is then

$$\mathcal{R}(\{\tilde{\gamma}, \gamma - \tilde{\gamma}\}, t_\gamma) = \frac{V_{BH}(\gamma, t_\gamma)}{(M(\tilde{\gamma}, t_\gamma, \Sigma) + |Z(\gamma - \tilde{\gamma}, t_\gamma)|)^2}. \tag{4.35}$$

The upperbound $M(\tilde{\gamma}, t, \Sigma)^2 \leq V_{BH}(\tilde{\gamma}, t)$ implies,

$$\mathcal{R}(\{\tilde{\gamma}, \gamma - \tilde{\gamma}\}, t_\gamma) \geq \frac{V_{BH}(\gamma, t_\gamma)}{(\sqrt{V_{BH}(\tilde{\gamma}, t_\gamma)} + |Z(\gamma - \tilde{\gamma}, t_\gamma)|)^2}. \tag{4.36}$$

Using the variables $x = \tilde{q}_0/q_0$ and $z = \tilde{p}/p$, we can evaluate the rhs exactly, such that

$$\mathcal{R}(\{\tilde{\gamma}, \gamma - \tilde{\gamma}\}, t_\gamma) \geq \frac{16}{(2\sqrt{3z^2 + x^2} + |3z - x - 2|)^2}. \tag{4.37}$$

We find that in case the BPS black hole has positive D0-brane charge, $q_0 - \tilde{q}_0 \geq 0$, the rhs is always $\leq 1$. However, if the D0-D4 state is polar, such that $q_0 - \tilde{q}_0 < 0$ the ratio is larger than 1, and thus provides a viable decay channel. This is similar to what we found for decay channel **4.2c**. It would be interesting to study such decay channels from a multi-center perspective as in [61]. We leave a more in depth analysis of such decay channels for future work.

## 4.4 Decay channels for CICYs with autochthonous divisors

Let us now turn to the complete intersection Calabi-Yau models with $h^{1,1}(X) = 2$ studied in Sec. 3.[9] This family of CY manifolds has $C_{111} = C_{112} = 0$. For this family, we can make contact with the recent investigations of five-dimensional solutions in [29, 30], which will be done in Sec. 5.

We start by considering the ratio of $V_{BH}/|Z|^2$. We find for the particular attractor solution, (3.7), that $V_{BH}(\gamma, t_\gamma) = 16\hat{q}_0^2$, and we have the ratio

$$\frac{V_{BH}(\gamma, t_\gamma)}{|Z(\gamma, t_\gamma)|^2} = 4. \tag{4.38}$$

This equals the ratio for the general attractor solution (3.2).

**Channel 4.4a: Charges spanned by $D_1$ and $D_2$**
We proceed by considering decay channels for vanishing electric charge, $q_a = 0$. Thus the total charge reads $\gamma = q_0 + p^1 D_1 + p^2 D_2$ with $D_1$ and $D_2$ divisors. The constraints on the charges for this attractor point are $p^2 > 0$, $p^1 < -\frac{2C_{222}}{3C_{122}}p^2$ and $\hat{q}_0 > 0$. We can thus set $p^1 = -\frac{2C_{222}}{3C_{122}}np^2$ for some $n > 1$. For very large $n$, this charge approaches the cone of charges populated by BPS black holes.

For the decay channels, we first study

$$\begin{aligned} \gamma_0 &= q_0, \\ \gamma_1 &= p^1 D_1, \\ \gamma_2 &= p^2 D_2, \end{aligned} \tag{4.39}$$

such that $\gamma = \gamma_0 + \gamma_1 + \gamma_2$.

In contrast to the general solution we now find

$$\begin{aligned} \mathcal{R}(\gamma, \{\gamma_0, \gamma_1, \gamma_2\}) &= \frac{V_{BH}(\gamma, t_\gamma)}{(|Z(\gamma_0, t_\gamma)| + |Z(\gamma_1, t_\gamma)| + |Z(\gamma_2, t_\gamma)|)^2} \\ &= \frac{4(1 - 2n)^2}{(1 - 4n)^2} < 1. \end{aligned} \tag{4.40}$$

This ratio is shown in Fig. 1, and we see that it asymptotes to one for large $n$. This is natural to expect, since in the large-$n$ limit the particular solution becomes the

---

[9]The story is analogous for the THCY of Table 3.

BPS one, with a different sign on $p^1$. However, in general we thus see that, we do not have a threshold bound state as for the generic solution. It is therefore interesting to see how this non-BPS state can decay.

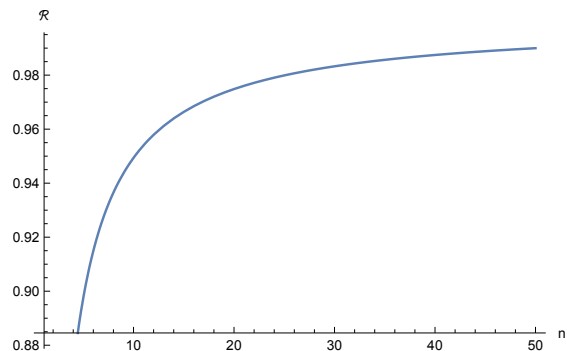

**Figure 1**. Mass ratio of Eq. (4.40) as function of the proportionality constant $n$ between the magnetic charges.

### Channels 4.4b: Charges spanned by $D_1, D_2$ and the autochthonous divisor $D_3$

The charges of the constituents above contained positive linear combinations of $D_1$ and $D_2$. As mentioned in Section 3, the effective cone is generated by an extra, or autochthonous, effective divisor. This is of the general form $D_3 = mD_2 - D_1$, for some $m \geq 1$. We have listed these divisors for various Calabi-Yaus in Appendix B. Since the cone extends beyond that generated by $D_1$ and $D_2$, we should allow for BPS constituents with magnetic charge $\tilde{p}^1 D_1 + \tilde{p}^3 D_3$ with $\tilde{p}_1, \tilde{p}_3 \geq 0$.

We will study this in what follows. We keep the charge ratio $p^1 = -\frac{2C_{222}}{3C_{122}} np^2$ with $n > 1$, as before. For the decay channels, we now consider four constituents with three carrying a magnetic charge,

$$
\begin{aligned}
\gamma_0 &= x_0 q_0, \\
\gamma_1 &= x_1 q_0 + \tilde{p}^1 D_1, \\
\gamma_2 &= x_2 q_0 + \tilde{p}^2 D_2, \\
\gamma_3 &= x_3 q_0 + \tilde{p}^3 D_3 = x_3 q_0 + \tilde{p}^3 (mD_2 - D_1).
\end{aligned}
\tag{4.41}
$$

From charge conservation we have the following restrictions that must be satisfied

$$
\sum_j x_j = 1, \qquad p^1 = \tilde{p}^1 - \tilde{p}^3, \qquad p^2 = \tilde{p}^2 + m\tilde{p}^3.
\tag{4.42}
$$

We can further assume that $\tilde{p}^3 = z_3 p^2$ for some rational number $z_3$. We now find

$$
\begin{aligned}
Z_0 &= x_0 q_0, \\
Z_1 &= q_0 \left( x_1 + 3\frac{\frac{C_{122}}{C_{222}}}{2n-1} \left( z_3 - \frac{2C_{222}}{3C_{122}} n \right) \right), \\
Z_2 &= q_0 \left( x_2 + \frac{4n-1}{2n-1}(1 - mz_3) \right), \\
Z_3 &= q_0 \left( x_3 + \frac{z_3}{2n-1} \left( m(4n-1) - 3\frac{C_{122}}{C_{222}} \right) \right).
\end{aligned}
\tag{4.43}
$$

To have BPS or anti-BPS constituents, it is important that the signs of the coefficients in front of the charges are compatible, in the sense that we must have

$$
\begin{aligned}
\mathrm{sgn}(x_1) &= \mathrm{sgn}(\tilde{p}^1) = \mathrm{sgn}\left( z_3 - \frac{2C_{222}}{3C_{122}} n \right), \\
\mathrm{sgn}(x_2) &= \mathrm{sgn}(\tilde{p}^2) = \mathrm{sgn}(1 - mz_3), \\
\mathrm{sgn}(x_3) &= \mathrm{sgn}(z_3).
\end{aligned}
\tag{4.44}
$$

Now, for brevity, let us consider one particular case, namely the K3 fibration #7887 in Table 2, also studied in [29]. This has $C_{122} = 4$, $C_{222} = 2$ and $D_3 = 4D_2 - D_1$. This means that we now have $p^1 = -\frac{n}{3}p^2$. Taking the above analysis and constraints into consideration, and assuming that $x_0 \geq 0$,[10] we end up with four different expressions for the mass ratios, viable in four different regimes for the charges:

- For $x_1 \geq 0$, $x_2 \leq 0$ $z_3 \geq n/3$:

$$
\mathcal{R}_1(\gamma, \{\gamma_j\}) = \frac{4(2n-1)^2}{(x_2 - 2n(1 + x_2 - 8z_3) - 4z_3)^2}.
$$

- For $x_1 \leq 0$, $x_2 \leq 0$, $1/4 \leq z_3 \leq n/3$:

$$
\mathcal{R}_2(\gamma, \{\gamma_j\}) = \frac{4(2n-1)^2}{((2n-1)(x_1 + x_2) - 2(8n-5)z_3)^2}.
$$

- For $x_1 \leq 0$, $x_2 \geq 0$, $0 \leq z_3 \leq 1/4$:

$$
\mathcal{R}_3(\gamma, \{\gamma_j\}) = \frac{4(2n-1)^2}{(1 + 2n(x_1 - 2) - x_1 + 6z_3)^2}.
$$

- For $x_1 \leq 0$, $x_2 \geq 0$, $z_3 \leq 0$:

$$
\mathcal{R}_4(\gamma, \{\gamma_j\}) = \frac{4(2n-1)^2}{(x_0 + x_2 - 2n(1 + x_0 + x_2 - 8z_3) - 4z_3)^2}.
$$

---

[10]similar results are found when assuming $x_0 \leq 0$

We can maximise these ratios separately over their corresponding domains, and its easy to show that the maximising values for $x_0$, $x_1$, $x_2$ and $z_3$ are given by

$$
\begin{aligned}
\mathcal{R}_1: & \quad x_0 = 1/4,\ x_1 = 1/2,\ x_2 = 0,\ z_3 = n/3, \\
\mathcal{R}_2: & \quad x_0 = 1/2,\ x_1 = 0,\ x_2 = 0,\ z_3 = 1/4, \\
\mathcal{R}_3: & \quad x_0 = 1/4,\ x_1 = 0,\ x_2 = 1/2,\ z_3 = 1/4, \\
\mathcal{R}_4: & \quad x_0 = 1/2,\ x_1 = 0,\ x_2 = 1/2,\ z_3 = 0.
\end{aligned}
\tag{4.45}
$$

The resulting maximised ratios are then functions of the proportionality constant $n$ between $p^1$ and $p^2$, and are shown in Fig. 2. We can clearly see that the maximum values for $\mathcal{R}_2$ and $\mathcal{R}_3$, which are both given by

$$
\mathcal{R}_2^{\max} = \mathcal{R}_3^{\max} = \frac{16(1-2n)^2}{(5-8n)^2},
\tag{4.46}
$$

allows for a ratio larger than unity for any $n$ (although tending to one for large $n$, for the same reason as mentioned above). This means that these regimes allow for the decay of the non-supersymmetric black hole.

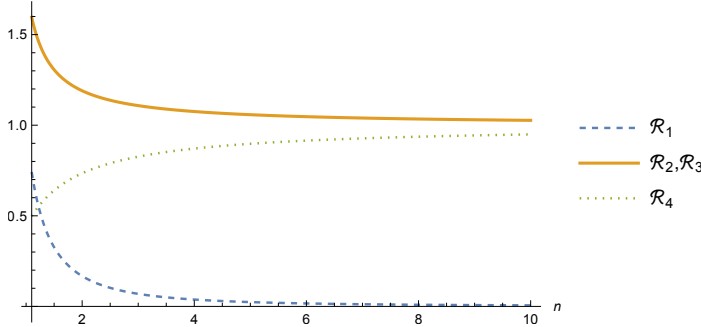

**Figure 2**. Maximised mass ratios for the K3 fibration, or 7887 of Tab. 2, as functions of the proportionality constant $n$. The two ratios $\mathcal{R}_2$ and $\mathcal{R}_3$ coincide.

We can do the same analysis for the other CICYs of Table 2, the result is that 11 out of the 16 gives exactly the same behaviour, i.e. exactly the same maximized ratios as above while the remaining five (7817, 7840, 7858, 7873 and 7885 in Table 2) differ slightly. As an example, the results for the CICY 7817 is shown in Fig. 3. Interestingly, we see that for $n$ sufficiently close to 1 we now have two distinct decay channels for the non-supersymmetric black hole. This happens for all the five CICYs not giving the same behaviour as Fig. 2.

We can see that the autochthonous divisors are vital for decay at tree level. If we consider only the cone generated by positive linear combinations of $D_1$ and $D_2$, we have that the most general BPS-anti-BPS decay products are given by

$$
\begin{aligned}
\gamma_1 &= x q_0 - z_1 p^1 D_1 + z_2 p^2 D_2, \\
\gamma_2 &= (1-x) q_0 + (1+z_1) p^1 D_1 + (1-z_2) p^2 D_2,
\end{aligned}
\tag{4.47}
$$

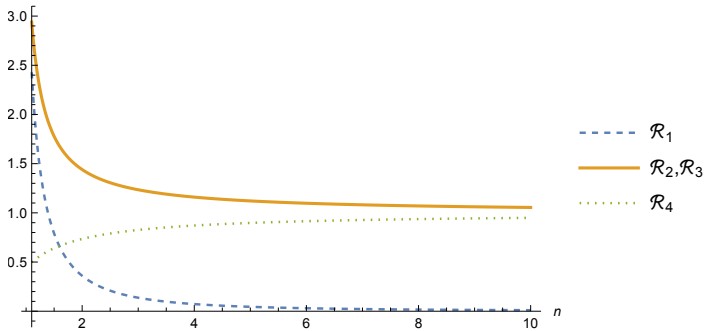

**Figure 3**. Maximised mass ratios for the CICY 7817 of Table 2 as functions of the proportionality constant $n$. The two ratios $\mathcal{R}_2$ and $\mathcal{R}_3$ coincide completely.

for some numbers $x, z_2 \geq 1$ and $z_1 \geq 0$. The mass ratio (4.2) is then maximised by the saturating values, $x = z_2 = 1$, $z_1 = 0$, giving the result (4.40). From the graphs we also observe that for large $n$ the ratios approach 1 (or less), which is consistent since for increasing $n$ the charge of the black hole approaches the cone of charges of BPS black holes.

The story is different for the general solutions (3.2). Since the charge of the D0 brane is negative for the general non-BPS solution we think of this as an anti-D0 brane. We consider the same four constituents as before, (4.41). Performing the same analysis as above, dividing the problem into domains depending on the signs of the coefficients and maximising the ratio for each case, we now find that the maximum values are always equal to the threshold value of one.

## 4.5 A model without autochthonous divisors

As discussed in Sec. 3.4 there are three two-moduli Calabi-Yau in [51] without an autochthonous divisor. These all have $C_{111} = C_{222} = 0$. They are the bi-cubic in $\mathbb{P}^2 \times \mathbb{P}^2$ with $C_{112} = C_{122} = 3$ in [29] and two THCY with $C_{112} = C_{122} = 1$ or 3 respectively in [30]. Refs [29, 30] found that non-BPS 5-dimensional black strings are valid and stable solutions against decay for $p^1/p^2 < 0$. This gives rise to recombination of holomorphic and anti-holomorphic cycles mentioned before. We will discuss in this subsection, non-BPS black holes for these geometries.

Since there are no autochthonous divisors generating a large cone, the effective cones of these CYs are simply generated by $D_1$ and $D_2$, the pull backs of generators of the effective cone of the ambient space. As in Sec. 3, we focus on one example with $C_{112} = C_{122} = 1$ and $p^2 = -4p^1$. From the Kähler condition we must have that $\hat{q}_0$ and $p^1$ have the same sign. We thus take $p^1 > 0$, which means that $p^2 < 0$. For the decay channels we consider decay into a BPS and anti-BPS constituent,

$$\begin{aligned}
\gamma_1 &= xq_0 + z_1 p^1 D_1 - z_2 p^2 D_2, \\
\gamma_2 &= (1 - x)q_0 + (1 - z_1)p^1 D_1 + (1 + z_2)p^2 D_2,
\end{aligned} \qquad (4.48)$$

for $x$, $z_1 \geq 1$ and $z_2 \geq 0$. The mass ratio is then however always smaller than one, and is maximised by the saturation values $x = z_1 = 1$, $z_2 = 0$, giving $\mathcal{R}(\gamma, \{\gamma_1, \gamma_2\}) \sim 0.83$. Including the possibility of decay into small BPS black holes with slight negative D0 charge, as in (4.23), does not seem to improve drastically on this bound, and it does not reach above unity after taking this into consideration. Here we further note that the maximum values are given when the $C$'s of each constituents are equal to zero such that for $q_0$ to be smaller than zero as in (4.23) we need to consider the $c_{2,a}$ corrections in $c_L$.

Thus also in this 4-dimensional case, this example suggests that the spectrum contains a stable non-BPS object. The WGC suggests that these only have small charges. It would be worthwhile to study more potential decay channels and to properly include $R^2$ corrections to the mass of the non-BPS solutions.

## 5   Lifting to M-theory

The 4-dimensional theory we have considered can be seen as M-theory on a CY3 $X$ times a circle. From the M-theory perspective the charges of the D2 and D4 branes then comes from M2 and M5 branes wrapping cycles in the CY while the D0 brane charge corresponds to momentum along the circle. If we decompactify the circle we get five-dimensional theories as studied in [29, 30]. Let us therefore briefly discuss how to relate the solutions.

In five dimensions the overall volume of $X$ is not dynamical, so we have one less degree of freedom to work with. This effectively means that the one-parameter Calabi-Yau manifolds will not give a dynamical theory in 5d. We therefore restrict to the case of $h^{1,1}(X) = 2$. In five dimensions we will also have that electric objects are pointlike, i.e. correspond to black holes, while magnetic objects are string like, i.e, correspond to black strings. So they are treated separately in [29, 30]. We will focus on discussing the black strings, as these are related to the dyonic solutions we have studied in this paper.

Let us denote the five-dimensional vector multiplet (real) scalars by $\tau^a$ and fix the overall (tree level) volume of the five-dimensional moduli space,

$$V_{5d} = \frac{1}{6} C_{abc} \tau^a \tau^b \tau^c, \tag{5.1}$$

to be equal to one. For the case $C_{111} = C_{112} = 0$, it was found in [29, 30] that the five-dimensional black string attractor solutions are[11]

$$\text{BPS:} \begin{cases} \tau_\gamma^1 = & \frac{p^1}{(C/6)^{1/3}}, \\ \tau_\gamma^2 = & \frac{p^2}{(C/6)^{1/3}}, \end{cases} \qquad \text{non-BPS:} \begin{cases} \tau_\gamma^1 = & -\frac{3C_{122}p^1 + 2C_{222}p^2}{3C_{122}(C/6)^{1/3}}, \\ \tau_\gamma^2 = & \frac{p^2}{(C/6)^{1/3}}. \end{cases} \tag{5.2}$$

---

[11]Note our conventions differ from those of [30]. We have rescaled $C$ by a factor of $-1/6$.

In [29] the main example of this type is that of the K3-fibration having $C_{122} = 4$ and $C_{222} = 2$.

The relation between the four-dimensional Kähler moduli, $J^a$, arising from compactifying type IIA string theory on a CY3, and the five-dimensional vector multiplet scalars, $\tau^a$, coming from compactifying M-theory on the same CY3, is given by

$$\tau^a = \frac{1}{V_{IIA}^{1/3}} J^a, \tag{5.3}$$

where $V_{IIA}$ is, as before, the volume of $X$ in string units (2.7) [26, 56]. Given the four-dimensional attractor solutions of Sec. 3, it is straightforward to evaluate $V_{IIA}$ at the various attractor points,

$$V_{IIA} = \begin{cases} \sqrt{\frac{6q_0^3}{C}}, & \text{BPS (3.2)}, \\ \sqrt{-\frac{6q_0^3}{C}}, & \text{General non-BPS (3.2)} \\ \sqrt{-\frac{6q_0^3}{C}}, & \text{Particular non-BPS (3.7)}. \end{cases} \tag{5.4}$$

It is now straightforward to check that the 4d BPS solutions as well as the particular non-BPS solutions, (3.7), satisfy (5.3) when compared to (5.2). Namely, we have

$$\frac{J_\gamma^a}{\tau_\gamma^a} = \left| \frac{6q_0^3}{C} \right|^{1/6}, \tag{5.5}$$

while for the general non-BPS solution, (3.2), Eq. (5.3) only holds when compared with the 5d BPS solution. This is expected since the radius of the M-theory circle goes to infinity and the direction of the momentum does not break supersymmetry. We thus have only supersymmetric attractors at this point. Breaking supersymmetry through reversing of the orientation of the compactification manifold, such as we have done for the general solution by flipping the sign of $q_0$, is sometimes called "skew-whiffing" and appears in many places in the literature [57, 58].

In a similar way, we can also study the relation between the solutions for the models with $C_{111} = C_{222} = 0$ in four and five dimensions. For the solutions (3.10) we start by defining $x = \frac{J^1}{J^2}$ and then study the ratio

$$\frac{p^1}{p^2} = -\frac{x(2C_{112} + C_{122}x)(C_{122}^2 + C_{112}C_{122}x + 2C_{112}^2 x^2)}{(C_{122} + 2C_{112}x)(2C_{122}^2 + C_{112}C_{122}x + C_{112}^2 x^2)}. \tag{5.6}$$

This agrees with the corresponding ratios in the five-dimensional solutions of [30] when setting $C_{112} = C_{122} = 1$ or 3. So these should correspond to the same solutions. For the case of $C_{112} = C_{122} = 1$ and $p^2 = -4p^1$ we have the solutions (3.12) in four dimensions, while the corresponding non-supersymmetric solutions in five dimensions are

$$\tau_\gamma^1 = (-4 + \sqrt{17})^{1/3},$$
$$\tau_\gamma^2 = \left( \frac{7 + \sqrt{17}}{2} \right)^{1/3}. \tag{5.7}$$

It is again easy to calculate the 4d Kähler volume, $V_{IIA}$, at this attractor point and see that the relation (5.3) again hold for these solutions.

We have thus seen that the 4d particular solutions are the ones that lift to the 5d non-BPS black string solutions of [29, 30].

# 6    Discussion

Motivated by the weak gravity conjecture, we have studied double extremal attractor black holes in four-dimensional $\mathcal{N} = 2$ supergravity. In support of the conjecture, we have demonstrated many decay channels where decay of non-supersymmetric black holes into BPS and anti-BPS constituents is energetically favorable. An important aspect of our analysis is the attractor mechanism, which depends only on the extremality of the black holes and thus allows us to study both supersymmetric and non-supersymmetric solutions.

Eq. (4.23) demonstrates that for the general attractor points (3.2) at tree level, non-BPS extremal black holes can decay into D0-branes and "polar" D0-D4 branes. We have also explored higher derivative $R^2$ corrections from the vector multiplet sector [23, 24, 48, 49, 59]. Curiously, we find with Eq. (4.21) that these $R^2$ corrections make these decay channels more stable rather than unstable. This behavior is untypical for $R^2$ corrections, which commonly lead to a larger charge to mass ratio. Notable exceptions are identified for non-supersymmetric theories in [60, Section 2.2]. Since we studied in this paper supersymmetric theories, we expect that D-term higher derivative corrections will further correct the mass formula favoring decay of non-supersymmetric extremal black holes.

Our results are complementary to recent results on black strings in five-dimensional supergravity [29, 30]. Some qualitative differences between four- and five-dimensional supergravity is additional electric D0-brane charge, and that the B-field makes the moduli of the Calabi-Yau complex. In this paper we have discussed how these differences affect the results for black hole decay in four dimensions.

We conclude with mentioning a few directions which deserve further study:

1. It would be interesting to better understand the stability of black holes, whose decay channels to BPS and anti-BPS are only marginally unstable at tree level. A better understanding of the $R^2$ corrections to the threshold decay channels **4.2b** is desirable.

2. Studying the decay channels from the perspective of the 2-dimensional CFT could provide important insights in the decay processes.

3. It is desirable to carry out an analogous analysis for extremal black holes which are not double extremal, that is to say with a non-trivial flow for the moduli from spatial infinity to the horizon. To this end, one would need to understand

| Non-BPS solution | Decay | Mass ratio | Section |
|---|---|---|---|
| Gen. sol. w/ corr., (3.3) | D0s and D4s | $1 - \frac{3}{160}\frac{c_2\cdot p}{C} + \dots$ | 4.2b |
| Gen. sol. w/o corr., (3.2) | Polar D0s, and D4s | $1 + \frac{\tilde{q}_0}{2q_0} + \dots$ | 4.2c |
| Gen. sol. w/o corr., (3.2) | Various D0-D2-D4 systems | Always $\leq 1$ | 4.3a-4.3d |
| Gen. sol., (3.2) | BPS D0-D4, and non-BPS D0-D4 | Possibly $> 1$ | 4.3e |
| Part. sol., (3.7) | w/o considering autoch. divisor | $\frac{4(1-2n)^2}{(1-4n)^2} < 1$ | 4.4a |
| Part. sol., (3.7) | Including autoch. divisor | $\frac{16(1-2n)^2}{(5-8n)^2} > 1$ | 4.4b |
| Part. sol., (3.12) | No autoch. divisor | $< 1$ | 4.5 |

**Table 1**. We collect the various results on the decay channels for non-supersymmetric black holes considered in this paper. In Section 4.2b we consider the decay of the general solution, (3.3), into D0- and D4-branes when including $R^2$ corrections, while Secs. 4.2c and 4.3a-4.3e consider the decay into various D0-D2-D4 and anti-D0-D2-D4 systems without considering $R^2$ corrections. In Sections 4.4a and 4.4b we consider the decay of the particular solutions (3.7) into D0-D4 and anti-D0-D4 states whose magnetic charges are spanned either by the divisors inherited directly from the ambient space or by considering the extra autochthonous divisors. For decay to be possible we find that the autochthonous divisor must be considered. Finally, in Sec. 4.5 we consider Calabi-Yau manifolds that do not have such an autochthounous divisor, we then find that the mass ratio is always smaller than 1 suggesting that these non-BPS states are stable against decay.

the non-BPS attractor flows better, possibly including multi-centers [61]. This is also of interest for point 1 above, since one can then explore decay channels with non-BPS constituents, which may be more favorable than the ones with BPS and anti-BPS constituents. We have briefly explored this type of decay in channel **4.3e**.

Some possible avenues for progress in this direction is the use of "fake" supersymmetry [19], as well as the results of [62] and the solutions of [63].

## Acknowledgments

JA would like to thank Cody Long for explaining certain aspects of their paper [29]. We are also happy to thank Nima Arkani-Hamed, Matthew Rochford and Antoine Vincenti for discussions. The majority of this work was carried out while JA was a graduate student in the School of Mathematics, Trinity College Dublin. During this time, JA was supported by the Government of Ireland Postgraduate Scholarship Programme GOIPG/2020/910 of the Irish Research Council. JM is supported by the Laureate Award 15175 "Modularity in Quantum Field Theory and Gravity" of the Irish Research Council, and the Ambrose Monell Foundation.

# A Some useful formulas and notations

Using the notations introduced in Sec. 3 for the D0-D2-D4 system at tree level, with gauge $X^0 = 1$, we can list various useful formulas. First, we have for $p^0 = 0$,

$$\begin{aligned}
\partial_a W &= q_a - C_{abc} p^b t^c, \\
\partial_a K &= \frac{3i}{2} \frac{L_a}{L},
\end{aligned} \tag{A.1}$$

such that

$$\begin{aligned}
\nabla_a W &= q_a - C_{abc} p^b t^c + \frac{3i L_a}{2L} W, \\
(\nabla_a W)^* &= q_a - C_{abc} p^b \bar{t}^c - \frac{3i L_a}{L} W^*.
\end{aligned} \tag{A.2}$$

We also need the metric and its inverse

$$\begin{aligned}
g_{a\bar{b}} &= \frac{3}{4L} \left( \frac{3}{L} L_a L_b - 2 L_{ab} \right), \\
g^{a\bar{b}} &= \frac{2L}{3} \left( \frac{3}{L} J^a J^b - L^{ab} \right),
\end{aligned} \tag{A.3}$$

where $L^{ab} L_{bc} = \delta^a{}_c$. From this we also find the Christoffel symbols

$$\Gamma^a_{bc} = \frac{3i}{2L} \left( L_b \delta^a_c + L_c \delta^a_b - L_{bc} J^a \right) - \frac{i}{2} L^{ad} C_{dbc}, \tag{A.4}$$

that appear in the equations of motion for the scalar moduli (2.22).

As mentioned in Sec. 3, at tree level we can express the superpotential as

$$W(\gamma) = q_0 + t^a q_a - \frac{1}{2} C_{abc} p^a t^b t^c = \hat{q}_0 - \frac{1}{2} C_{ab} \hat{t}^a \hat{t}^b, \tag{A.5}$$

where $\gamma = (q_0, q_a, p^a, 0)$ [17]. Since $t^a = B^a + iJ^a$ we can also write this as

$$W(\gamma) = q_0 + (B^a + iJ^a) q_a - \frac{1}{2} C_{ab} (B^a B^b + 2i J^a B^b - J^a J^b). \tag{A.6}$$

The real and imaginary parts of $W$ are then

$$\begin{aligned}
\mathrm{Re}(W) &= q_0 + B^a q_a - \frac{1}{2}(B \cdot B - J \cdot J) = \hat{q}_0 + \frac{1}{2}(J \cdot J - \hat{B} \cdot \hat{B}), \\
\mathrm{Im}(W) &= J^a q_a - J \cdot B = -J \cdot \hat{B}.
\end{aligned} \tag{A.7}$$

This means that we now have

$$\begin{aligned}
|W|^2 =& \frac{1}{4}(J \cdot J)^2 + (J \cdot J)\left( q_0 + B^a q_a - \frac{1}{2} B \cdot B \right) + J^a J^b q_a q_b + (J \cdot B)^2 - 2 J^a q_a (J \cdot B) \\
& + q_0^2 + B^a B^b q_a q_b + \frac{1}{4}(B \cdot B)^2 - (B \cdot B)(q_0 + B^a q_a) + 2 q_0 B^a q_a \\
=& \frac{1}{4}(J \cdot J)^2 + (J \cdot J)(\hat{q}_0 - \frac{1}{2}(\hat{B} \cdot \hat{B})) + (J \cdot \hat{B})^2 + \hat{q}_0^2 - \hat{q}_0(\hat{B} \cdot \hat{B}) + \frac{1}{4}(\hat{B} \cdot \hat{B})^2.
\end{aligned} \tag{A.8}$$

The black hole potential can be expressed in a similar way as

$$
\begin{aligned}
e^{-K}V_{BH} = &- 4V_{IIA}L^{ab}(q_a - C_{ac}B^c)(q_b - C_{bd}B^d) \\
&+ (J \cdot J)^2 - 4V_{IIA}(p \cdot J) + 2(J \cdot B)^2 + 2J^a J^b q_a q_b - 4J^a q_a(J \cdot B) \\
&+ 4q_0^2 + 4B^a B^b q_a q_b - 4(q_0 + B^a q_a)(B \cdot B) + (B \cdot B)^2 + 8q_0 B^a q_a \quad \text{(A.9)} \\
= &- 4V_{IIA}L^{ab}C_{ac}C_{bd}\hat{B}^c\hat{B}^d + (J \cdot J)^2 - 4V_{IIA}(p \cdot J) \\
&+ 2(J \cdot \hat{B})^2 + 4\hat{q}_0^2 - 4\hat{q}_0(\hat{B} \cdot \hat{B}) + (\hat{B} \cdot \hat{B})^2.
\end{aligned}
$$

We can simplify this expression further in special cases. First of all we consider a D0-D4 system, i.e. setting $q_a = 0$. This is equivalent to removing the hats in the above expression, i.e.,

$$
\begin{aligned}
e^{-K}V_{BH} = &(J \cdot J)^2 - 4V_{IIA}(p \cdot J) + 2(J \cdot B)^2 - 4V_{IIA}L^{ab}C_{ac}C_{bd}B^c B^d \\
&+ 4q_0^2 - 4q_0(B \cdot B) + (B \cdot B)^2.
\end{aligned} \quad \text{(A.10)}
$$

Alternatively, we can set $\hat{B}^a = 0$, which gives

$$
e^{-K}V_{BH} = (J \cdot J)^2 - 4V_{IIA}(p \cdot J) + 4\hat{q}_0^2. \quad \text{(A.11)}
$$

For the one moduli case, $h^{1,1}(X) = 1$, it is straightforward to determine the inverse $L^{ab}$. To state the result for this case, we introduce the shorthand notation $C_{111} = \kappa$, $p^1 = p$ and similarly for $J$ and $B$. This gives now

$$
e^{-K}V_{BH} = \frac{\kappa^2 p^2}{3}\left(J^4 + 4\hat{B}^2 J^2 + 3\hat{B}^4 - \frac{12\hat{q}_0}{\kappa p}\hat{B}^2 + \frac{12\hat{q}_0^2}{\kappa^2 p^2}\right). \quad \text{(A.12)}
$$

We can also note that $L_{ab}$ is a quadratic form on $H^2(X, \mathbb{R})$ with the signature $(1, b_2 - 1)$, similar to $C_{ab}$.

# B  Data of Calabi-Yau manifolds

In this Appendix, we collect some data of the Calabi-Yau families of interest to this paper. This data is collected in the Tables 2-4. The presentation follows that of [51].

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

| # | $(C_{122}, C_{222})$ | $n$ | Generators |
|---|---|---|---|
| 7806 | (6,6) | 4 | $D_1, 2D_2 - D_1$ |
| 7816 | (8,8) | 5 | $D_1, 2D_2 - D_1$ |
| 7817 | (8,12) | 6 | $D_1, D_2 - D_1$ |
| 7819 | (8,16) | 7 | $D_1, D_2 - D_1$ |
| 7822 | (8,8) | 5 | $D_1, 2D_2 - D_1$ |
| 7823 | (8,16) | 6 | $D_1, D_2 - D_1$ |
| 7840 | (6,9) | 5 | $D_1, D_2 - D_1$ |
| 7858 | (6,5) | 4 | $D_1, 2D_2 - D_1$ |
| 7867 | (6,12) | 6 | $D_1, D_2 - D_1$ |
| 7869 | (6,12) | 5 | $D_1, D_2 - D_1$ |
| 7873 | (6,8) | 5 | $D_1, D_2 - D_1$ |
| 7882 | (6,4) | 4 | $D_1, 3D_2 - D_1$ |
| 7885 | (4,5) | 4 | $D_1, D_2 - D_1$ |
| 7886 | (4,8) | 5 | $D_1, D_2 - D_1$ |
| 7887 | (4,2) | 3 | $D_1, 4D_2 - D_1$ |
| 7888 | (4,8) | 4 | $D_1, D_2 - D_1$ |

**Table 2**. Relevant data for the 16 complete intersection Calabi-Yau threefolds with $C_{111} = C_{112} = 0$ taken from [51]. The number in the first column are the names given in [51]. The ambient space is of the form $\mathcal{A} = \mathbb{P}^1 \times \mathbb{P}^n$, and the last column lists the generators of the effective cone.

[4] H. Ooguri and C. Vafa, *Non-supersymmetric AdS and the Swampland, Adv. Theor. Math. Phys.* **21** (2017) 1787–1801, [1610.01533].

[5] Y. Kats, L. Motl and M. Padi, *Higher-order corrections to mass-charge relation of extremal black holes, JHEP* **12** (2007) 068, [hep-th/0606100].

[6] C. Cheung and G. N. Remmen, *Naturalness and the Weak Gravity Conjecture, Phys. Rev. Lett.* **113** (2014) 051601, [1402.2287].

[7] C. Cheung, J. Liu and G. N. Remmen, *Proof of the Weak Gravity Conjecture from Black Hole Entropy, JHEP* **10** (2018) 004, [1801.08546].

| # | $(C_{111}, C_{112}, C_{122}, C_{222})$ | Generators |
|---|---|---|
| (7,1) | (8,4,0,0) | $D_2$, $D_1 - 2D_2$ |
| (8,1) | (8,4,0,0) | $D_2$, $D_1 - D_2$ |
| (8,2) | (8,4,0,0) | $D_2$, $D_1 - D_2$ |
| (9,1) | (0,0,4,2) | $D_1$, $4D_2 - D_1$ |
| (13,1) | (4,5,0,0) | $D_2$, $D_1 - D_2$ |
| (28,1) | (4,2,0,0) | $D_2$, $D_1 - 2D_2$ |
| (29,1) | (4,2,0,0) | $D_2$, $D_1 - D_2$ |
| (29,2) | (4,2,0,0) | $D_2$, $D_1 - D_2$ |
| (30,1) | (108,12,0,0) | ? |
| (32,2) | (108,12,0,0) | ? |

**Table 3**. Relevant data for the 10 toric hypersurace Calabi-Yau threefolds with either $C_{122} = C_{222} = 0$ or $C_{111} = C_{112} = 0$ taken from [51]. The numbers in the first column are the names given in [51]. The last column lists the generators of the effective cone, where for the last two manifolds the generators of the effective cone are not found in [51].

| # | $(C_{112}, C_{122})$ | Type |
|---|---|---|
| 7884 | (3,3) | CICY, $\mathcal{A} = \mathbb{P}^2 \times \mathbb{P}^2$ |
| (1,1) | (1,1) | THCY |
| (5,2) | (3,3) | THCY |

**Table 4**. Relevant data for the 3 Calabi-Yau threefolds with $C_{111} = C_{222} = 0$ from [51]. The numbers in the first column are the names given in [51].

[8] L. Aalsma, A. Cole and G. Shiu, *Weak Gravity Conjecture, Black Hole Entropy, and Modular Invariance*, *JHEP* **08** (2019) 022, [1905.06956].

[9] S.-J. Lee, W. Lerche and T. Weigand, *Tensionless Strings and the Weak Gravity Conjecture*, *JHEP* **10** (2018) 164, [1808.05958].

[10] S.-J. Lee, W. Lerche and T. Weigand, *A Stringy Test of the Scalar Weak Gravity Conjecture*, *Nucl. Phys. B* **938** (2019) 321–350, [1810.05169].

[11] N. Gendler and I. Valenzuela, *Merging the Weak Gravity and Distance Conjectures*

*Using BPS Extremal Black Holes*, 2004.10768.

[12] B. Heidenreich, M. Reece and T. Rudelius, *Repulsive Forces and the Weak Gravity Conjecture*, *JHEP* **10** (2019) 055, [1906.02206].

[13] A. M. Charles, *The Weak Gravity Conjecture, RG Flows, and Supersymmetry*, 1906.07734.

[14] S. Ferrara, R. Kallosh and A. Strominger, *N=2 extremal black holes*, *Phys. Rev. D* **52** (1995) R5412–R5416, [hep-th/9508072].

[15] S. Ferrara and R. Kallosh, *Supersymmetry and attractors*, *Phys. Rev. D* **54** (1996) 1514–1524, [hep-th/9602136].

[16] T. Mohaupt, *Black hole entropy, special geometry and strings*, *Fortsch. Phys.* **49** (2001) 3–161, [hep-th/0007195].

[17] P. K. Tripathy and S. P. Trivedi, *Non-supersymmetric attractors in string theory*, *JHEP* **03** (2006) 022, [hep-th/0511117].

[18] R. Kallosh, N. Sivanandam and M. Soroush, *The Non-BPS black hole attractor equation*, *JHEP* **03** (2006) 060, [hep-th/0602005].

[19] A. Ceresole and G. Dall'Agata, *Flow Equations for Non-BPS Extremal Black Holes*, *JHEP* **03** (2007) 110, [hep-th/0702088].

[20] J. M. Maldacena, A. Strominger and E. Witten, *Black hole entropy in M theory*, *JHEP* **12** (1997) 002, [hep-th/9711053].

[21] R. Minasian, G. W. Moore and D. Tsimpis, *Calabi-Yau black holes and (0,4) sigma models*, Commun. Math. Phys. **209**, 325-352 (2000) [arXiv:hep-th/9904217 [hep-th]].

[22] A. Dabholkar, A. Sen and S. P. Trivedi, *Black hole microstates and attractor without supersymmetry*, *JHEP* **01** (2007) 096, [hep-th/0611143].

[23] P. Kraus and F. Larsen, *Microscopic black hole entropy in theories with higher derivatives*, *JHEP* **09** (2005) 034, [hep-th/0506176].

[24] P. Kraus and F. Larsen, *Holographic gravitational anomalies*, *JHEP* **01** (2006) 022, [hep-th/0508218].

[25] F. Denef and G. W. Moore, *Split states, entropy enigmas, holes and halos*, *JHEP* **11** (2011) 129, [hep-th/0702146].

[26] J. de Boer, F. Denef, S. El-Showk, I. Messamah and D. Van den Bleeken, *Black hole bound states in AdS(3) x S**2*, *JHEP* **11** (2008) 050, [0802.2257].

[27] G. W. Gibbons, *Vacuum Polarization and the Spontaneous Loss of Charge by Black Holes*, Commun. Math. Phys. **44** (1975), 245-264 doi:10.1007/BF01609829

[28] W. A. Hiscock and L. D. Weems, *Evolution of Charged Evaporating Black Holes*, Phys. Rev. D **41** (1990), 1142 doi:10.1103/PhysRevD.41.1142

[29] C. Long, A. Sheshmani, C. Vafa and S.-T. Yau, *Non-Holomorphic Cycles and Non-BPS Black Branes*, 2104.06420.

[30] A. Marrani, A. Mishra and P. K. Tripathy, *Non-BPS Black Branes in M-theory over Calabi-Yau Threefolds*, 2202.06872.

[31] M. Demirtas, C. Long, L. McAllister and M. Stillman, *Minimal Surfaces and Weak Gravity*, *JHEP* **03** (2020) 021, [1906.08262].

[32] F. Denef, *Supergravity flows and D-brane stability*, *JHEP* **08** (2000) 050, [hep-th/0005049].

[33] S. Ferrara, G. W. Gibbons and R. Kallosh, *Black holes and critical points in moduli space*, *Nucl. Phys. B* **500** (1997) 75–93, [hep-th/9702103].

[34] G. W. Gibbons, R. Kallosh and B. Kol, *Moduli, scalar charges, and the first law of black hole thermodynamics*, *Phys. Rev. Lett.* **77** (1996) 4992–4995, [hep-th/9607108].

[35] M. Shmakova, *Calabi-Yau black holes*, *Phys. Rev. D* **56** (1997) 540–544, [hep-th/9612076].

[36] G. W. Moore, *Arithmetic and attractors*, hep-th/9807087.

[37] P. Candelas, X. de la Ossa, M. Elmi and D. Van Straten, *A One Parameter Family of Calabi-Yau Manifolds with Attractor Points of Rank Two*, *JHEP* **10** (2020) 202, [1912.06146].

[38] K. Bönisch, A. Klemm, E. Scheidegger and D. Zagier, *D-brane masses at special fibres of hypergeometric families of Calabi-Yau threefolds, modular forms, and periods*, 2203.09426.

[39] P. Candelas, P. Kuusela and J. McGovern, *Attractors with large complex structure for one-parameter families of Calabi-Yau manifolds*, *JHEP* **11** (2021) 032, [2104.02718].

[40] K. Goldstein, N. Iizuka, R. P. Jena and S. P. Trivedi, *Non-supersymmetric attractors*, *Phys. Rev. D* **72** (2005) 124021, [hep-th/0507096].

[41] J. de Boer, M. C. N. Cheng, R. Dijkgraaf, J. Manschot and E. Verlinde, *A Farey Tail for Attractor Black Holes*, *JHEP* **11** (2006) 024, [hep-th/0608059].

[42] D. Gaiotto, A. Strominger and X. Yin, *The M5-Brane Elliptic Genus: Modularity and BPS States*, *JHEP* **08** (2007) 070, [hep-th/0607010].

[43] R. Dijkgraaf, J. M. Maldacena, G. W. Moore and E. P. Verlinde, *A Black hole Farey tail*, [hep-th/0005003].

[44] J. Manschot, *Stability and duality in N=2 supergravity*, *Commun. Math. Phys.* **299** (2010) 651–676, [0906.1767].

[45] S. Alexandrov, J. Manschot and B. Pioline, *D3-instantons, Mock Theta Series and Twistors*, *JHEP* **04** (2013) 002, [1207.1109].

[46] G. Lopes Cardoso, B. de Wit and T. Mohaupt, *Corrections to macroscopic supersymmetric black hole entropy*, *Phys. Lett. B* **451** (1999) 309–316, [hep-th/9812082].

[47] A. Sen, *Black hole entropy function and the attractor mechanism in higher derivative gravity*, *JHEP* **09** (2005) 038, [`hep-th/0506177`].

[48] B. Sahoo and A. Sen, *Higher derivative corrections to non-supersymmetric extremal black holes in N=2 supergravity*, *JHEP* **09** (2006) 029, [`hep-th/0603149`].

[49] G. L. Cardoso, B. de Wit and S. Mahapatra, *Black hole entropy functions and attractor equations*, *JHEP* **03** (2007) 085, [`hep-th/0612225`].

[50] B. de Wit, S. Katmadas and M. van Zalk, *New supersymmetric higher-derivative couplings: Full N= 2 superspace does not count!*, *JHEP***01** (2011), [`1010.2150`].

[51] C. R. Brodie, A. Constantin, A. Lukas and F. Ruehle, *Geodesics in the extended Kähler cone of Calabi-Yau threefolds*, `2108.10323`.

[52] J. Ottem, *Birational geometry of hypersurfaces in products of projective spaces*, *Mathematische Zeitschrift* **280** (05, 2013) .

[53] M. Demirtas, C. Long, L. McAllister and M. Stillman, *The Kreuzer-Skarke Axiverse*, *JHEP* **04** (2020) 138, [`1808.01282`].

[54] E. Gruss and Y. Oz, *Large Charge Four-Dimensional Extremal N=2 Black Holes with R**2 - Terms*, *JHEP* **05** (2009) 041, [`0902.0831`].

[55] E. G. Gimon, F. Larsen and J. Simon, *Constituent Model of Extremal non-BPS Black Holes*, *JHEP* **07** (2009) 052, [`0903.0719`].

[56] D. Gaiotto, A. Strominger and X. Yin, *5D black rings and 4D black holes*, *JHEP* **02** (2006) 023, [`hep-th/0504126`].

[57] M. J. Duff, B. E. W. Nilsson and C. N. Pope, *Kaluza-Klein Supergravity*, *Phys. Rept.* **130** (1986) 1–142.

[58] M. Berkooz and S.-J. Rey, *Non-supersymmetric stable vacua of m-theory*, *Journal of High Energy Physics* **1999** (1999) 014.

[59] K. Saraikin and C. Vafa, *Non-supersymmetric black holes and topological strings*, *Class. Quant. Grav.* **25** (2008) 095007, [`hep-th/0703214`].

[60] N. Arkani-Hamed, Y. t. Huang, J. Y. Liu and G. N. Remmen, *Causality, unitarity, and the weak gravity conjecture*, *JHEP* **03** (2022), [`2109.13937`].

[61] G. Bossard and S. Katmadas, *Non-BPS walls of marginal stability*, *JHEP* **10** (2013) 1–55, [`1309.3236`].

[62] D. Astefanesei, H. Nastase, H. Yavartanoo and S. Yun, *Moduli flow and non-supersymmetric AdS attractors*, *JHEP* **04** (2008) 074, [`0711.0036`].

[63] P. K. Tripathy, *New branches of non-supersymmetric attractors in N = 2 supergravity*, *Phys. Lett. B* **770** (2017) 182–185, [`1701.00368`].