# Peer review of "Decay channels for double extremal black holes in four dimensions"

_SciPost Physics_

## Round 1 · Referee Report · Anonymous · 2023-6-1

Report
The paper addresses the stability of extremal, non-supersymmetric black holes in Calabi-Yau compactifications of IIA string theory in light of the weak gravity conjecture. It is generally well written, the technical analysis appears to be sound. However, a non-negligible fraction of the manuscript is concerned with the review and collection of already known facts. Still, a sufficient amount of original results is presented. In my opinion, it deserves publication after a few points are addressed:
- The authors should make the impact of their results in the context of the weak gravity conjecture more clear. At the end of the introduction (section 1) it is claimed that some of the analyzed states are expected to be "stable against decay". However, to my understanding, it is not explained why the decay channels that are considered here are the only possible ones. On the contrary, at the end of section 4 it is said that "It would be worthwhile to study more potential decay channels and to properly include R2 corrections to the mass of the non-BPS solutions." In the discussion (section 6) the possible existence of stable extremal black hole states does not seem to be mentioned at all. Also, it appears to me that the effect of higher-derivative corrections is only taken into account for some of the cases in section 4. A more systematic discussion and interpretation of the results seems to be in order.
- The statement "It is important to note that the WGC is a statement about low-energy theories and therefore only macroscopic states need to satisfy the bound" in the paragraph below eq. (1.1) is a bit confusing. Normally, the weak gravity conjecture is formulated as the requirement that at least one state satisfies the bound (1.1). On the other hand, the existence of other (also macroscopic) states that violate this bound is explicitly allowed (and generally expected). Also, it is not entirely clear what is meant by "macroscopic" states in this context. This should be clarified (similarly for second to last paragraph of section 1).
- Below eq. (2.7) "the volume in 11D Planck units" is mentioned. It is not clear how this connects to the rest of the discussion in this section.
- At the beginning of section 2.3 it is stated that "It is clear from the above that the moduli ta(τ) are non-trivial functions of τ." This statement is of course generally correct but maybe slightly confusing in view of the preceding sentence "In this paper, we will only be concerned with decay channels for such double extremal black holes, either BPS or non-BPS" and the definition of "double extremal" black holes on the previous page.
- The definition of "polar states" below eq. (2.36) refers to "polar terms" in a partition function that has not been introduced previously. I don't think that this explanation is beneficial for the reader without providing further information. Also, I am not sure to which extend the previous discussion of the dual CFT is needed for the rest of the paper.

---

## Round 1 · Referee Report · Anonymous · 2023-6-29

Strengths
The paper discusses an important question in quantum gravity related to the weak gravity conjecture.
The obtained results are original and sound.
Weaknesses
The discussion of F-term higher derivative corrections is not fully justified.
Report
The paper describes possible decay channels for extremal non-BPS black holes to decay into BPS constituent in type IIA string theory on a Calabi-Yau threefold. Type II string theory on a Calabi-Yau threefold is one of the best supersymmetric models for studying quantum gravity questions in string theory. It is supersymmetric enough to include many exact formulas in a wide variety of backgrounds. The effective theory admits extremal non-BPS black holes that are classically stable, but are expected to decay into supersymmetric black holes through quantum effects. For this to be possible, one must first check the mass threshold for a non-BPS black hole to possibly decay into BPS constituents. The authors investigate this threshold bound in supergravity for a good number of explicit examples. They also discuss F-terms four-derivative corrections to the mass formula, although D-term corrections are a priori of the same order for non-supersymmetric black holes. The question addressed by the authors is very interesting and the paper provides non-trivial results. I believe it deserves to be published in scipost. There are nonetheless few details that could be clarified.
The denomination `decay channels' is slightly misleading since it suggests that the authors identify a quantum effect that would allow a non-BPS black hole to decay into BPS constituents. I do not know of such a result within the MSW CFT for example. The author only identify a possible state of BPS black holes with the same total charge and a lower total energy. It is not obvious that the decay is always possible in the quantum theory.
The points 1 page 3 of the introduction suggests instead that the analysis is carried out in the MSW CFT, whereas the threshold mass checks are done in supergravity. I believe the authors should state more explicitly in this point 1 that they only study the mass threshold in supergravity, and this is what they call decay channels.
Actually, reference 20 and 21 are also supergravity papers that only mention the MSW model as a side reference in the introduction. So maybe the author could clarify in a footnote how to deduce that non-BPS charges in the MSW model lie in "this cone of the charge lattice". If this is just to say that non-BPS black hole have the opposite D0 charge, this was already exhibited in the paper of Gimon-Larsen-Simon in 0710.4967.
At the end of 3.2 page 12, the authors mention the higher derivative corrections to Wald entropy. They only mention F-term higher derivative corrections, whereas there is no reason to expect that they are the only corrections to the non-BPS black hole's Wald entropy. Even for BPS black holes it was not that obvious, and was proved in JHEP 01 (2011), 007 [arXiv:1010.2150 [hep-th]].
For the point 3. in the conclusion, the decay channels of a non-supersymmetric black hole into a bound state of non-supersymmetric constituents was addressed in theories with a Jordan type intersection form in JHEP 10 (2013), 179 [arXiv:1309.3236 [hep-th]].
Despite the few queries I believe the paper well deserves to be published in scipost after minor revisions.
Requested changes
1) I believe the authors should state more explicitly in this point 1 that they only study the mass threshold in supergravity, and this is what they call decay channels.
2) The author should explain in the introduction that they do not study the non-BPS black holes in the MSW model .
3) I think it would help the geometry layman to make it clear in point 2 that the intersection number is the triple intersection tensor C_{abc} that determines the cubic prepotential in supergravity.
4) The ordering could be improved in section 2.4. The shifted D0 charge is used in page 9 whereas it is defined in page 10. Since the authors spend few paragraphs reviewing these definitions, they could maybe motivate that (2.39) is obtained by a fractional spectral flow such that one gets effectively a pure D0-D4 system.
5) The issue of D-terms is only mentioned at the end of page 20, whereas I believe it should appear more prominently. In particular in the conclusion when the authors write that "Curiously [...] R^2 corrections make these decay channels more stable rather than unstable." it could be good to say that this might just be a red herring due to the fact that the authors do not take the D-term corrections into account.
6) The results on the decay channels in various examples are scattered in the various subsections and it may be profitable to have a concluding table in the discussion section.

---

## Editorial Decision

unknown